# Integration of absolute multi-omics reveals dynamic protein-to-RNA ratios and metabolic interplay within mixed-domain microbiomes

F. Delogu [1✉], B. J. Kunath [1,2], P. N. Evans[3], M. Ø. Arntzen [1], T. R. Hvidsten [1] & P. B. Pope [1,4✉]

While the field of microbiology has adapted to the study of complex microbiomes via modern meta-omics techniques, we have not updated our basic knowledge regarding the quantitative levels of DNA, RNA and protein molecules within a microbial cell, which ultimately control cellular function. Here we report the temporal measurements of absolute RNA and protein levels per gene within a mixed bacterial-archaeal consortium. Our analysis of this data reveals an absolute protein-to-RNA ratio of $10^2$–$10^4$ for bacterial populations and $10^3$–$10^5$ for an archaeon, which is more comparable to Eukaryotic representatives' humans and yeast. Furthermore, we use the linearity between the metaproteome and metatranscriptome over time to identify core functional guilds, hence using a fundamental biological feature (i.e., RNA/protein levels) to highlight phenotypical complementarity. Our findings show that upgrading multi-omic toolkits with traditional absolute measurements unlocks the scaling of core biological questions to dynamic and complex microbiomes, creating a deeper insight into inter-organismal relationships that drive the greater community function.

[1] Faculty of Chemistry, Biotechnology and Food Science, Norwegian University of Life Sciences, Ås N-1432, Norway. [2] Luxembourg Centre for Systems Biomedicine, Université du Luxembourg, 4362 Esch-sur-Alzette, Luxembourg. [3] Australian Centre for Ecogenomics, School of Chemistry and Molecular Biosciences, University of Queensland, St Lucia, QLD, Australia. [4] Faculty of Biosciences, Norwegian University of Life Sciences, Ås N-1432, Norway. ✉email: fra.delogu92@gmail.com; phil.pope@nmbu.no

The fundamentals of microbiology have been built within the constrained framework of pure culture studies of model organisms that are grown under controlled steady-state conditions. However, we are constantly told that microorganisms grown in single culture behave in a different manner to those in mixed natural communities[1]. For example, when *Escherichia coli* is grown axenically in steady state, we can expect that each RNA molecule corresponds to $10^2$–$10^4$ of the matching protein (absolute protein-to-RNA ratio, hereafter referred protein-to-RNA ratio) and the variation in the level of cellular RNA explains ~29% of the variation in the amount of detectable protein[2]. Yet does this notion hold true when a given bacterial population is part of a larger community and subject to transitions from one state of equilibrium to another due to limiting and/or confronting environmental factors? In this context, the exploration of temporal interplay between populations with different lifestyles (comprising metabolism, motility, sporulation, etc.) becomes of primary importance to interpret the changes in fundamental quantities in a microbial community, such as the protein-to-RNA ratio that ultimately impacts the overarching community phenotype(s). In order to perform studies of such design and test if previously defined quantitative data about the functioning of microorganisms (i.e., protein-to-RNA ratio) is applicable to real world consortia, we must first sample microbial communities across transition events and employ quantification techniques that are absolute.

Meta-omics techniques, such as metagenomics (MG)[3,4], metatranscriptomics (MT)[5], and metaproteomics (MP)[6] are routinely used to assess microorganisms in the natural world, where they are part of communities that are frequently dominated by as-yet uncultivated populations[7]. The quantities retrieved from the meta-omics are usually expressed in relative terms, which makes a comparison between samples and between omic layers inaccurate[8,9]. Moreover, within dynamic data measurements, such as the MT or MP, the notion of steady state becomes relevant as it is extremely rare that parameters (e.g., bacterial growth rate and nutrient availability) are stable over time[9].

Here, we present an absolute temporal multi-omic analysis of a minimalistic cellulose-degrading and methane-producing consortium (SEM1b), which was resolved at the strain level and augmented with two strain isolates[10]. We combined both an RNA-spike-in for MT[11,12] and the *total protein approach* for MP[13] for the absolute quantification of high-throughput data. We not only demonstrate that temporal SEM1b samples were comparable within the same omic layer, but also between the MT and MP. Indeed, the protein-to-RNA ratio per sample of the bacterial populations matched previous calculations for the existing example from axenically cultured *E. coli*[2]. We present protein-to-RNA ratios for an archaeon (*Methanothermobacter thermoautotrophicus*), which are similar to those reported for the Eukarya, and support crystallography and homology studies that suggest the translation system of archaea more closely resembles eukaryotes[14]. Our approach enabled us to explore the linearity of the protein-to-RNA ratio and if it is influenced by changes in community state and/or specific population lifestyle. Finally, we estimated the translation and protein degradation rates, showing that a post-transcriptional down-regulation of protein levels marks main lifestyle changes (e.g., motility/chemotaxis and metabolism) during the community development.

## Results

**Taxonomic and functional resolution of the omics.** In order to explore the RNA/protein dynamics in a microbiome setting, we first needed to characterize our test community over time at the molecular level. We previously genomically reconstructed and resolved the SEM1b community, retrieving 11 metagenome-assembled genomes (MAGs) as well as two isolate genomes (see "Methods" section)[10], covering the taxonomic and functional niches that are required to convert cellulosic material to methane/$CO_2$ in an anaerobic biogas reactor[15]. Taxonomic analysis of both 16S rRNA genes and the MAGs of SEM1b inferred population-level affiliations to *Rumini(Clostridium) thermocellum* (RCLO1), *Clostridium* sp. (CLOS1), *Coprothermobacter proteolyticus* (COPR1, BWF2A, SW3C), *Tepidanaerobacter* (TEPI1-2), *Synergistales* (SYNG1-2), *Tissierellales* (TISS1), and the methanogen *Methanothermobacter* (METH1)[10], as depicted in Fig. 1a. Herein we estimated that the total genomic potential of SEM1b includes 39,144 open reading frames (ORFs) (Supplementary Data 1). Since ORFs with very high sequence similarity may produce RNAs and proteins that are indistinguishable in MT and MP data, all the ORFs were gathered into ORF-groups (ORFGs) during the MT and MP data processing (see "Methods" section), where a singleton ORFG is defined as a group with a single ORF, and thus a single gene. Using this approach, our MT and MP data identified 12552 (96% singleton) and 3235 (78% singletons) highly transcribed and translated ORFGs, respectively. The discrepancy between the singleton percentages was as expected, due to the fact that variations in DNA/RNA sequences are greater than in proteins since different codons can code for the same amino acid (codon degeneracy). Degeneracy also implies that the chance to distinguish between homologous genes using MT is greater than using MP. Previous MG analyses using assembly algorithms have shown that genomic regions difficult to assemble in a given environmental contig can harbor variants from multiple, closely related strains, which can be further linked to normal strain-level variability within a population and species divergence[16–18]. Within SEM1b, the ORFGs that contained multiple homologous ORFs predominantly originated from several strains of a single species. For example, in the MT, 444 non-singleton ORFGs (88% of the total) contained ORFs from different strains of the same species, whilst this was the case for 294 ORFGs (32%) in the MP.

All ORFs were annotated using KEGG Ontology (KO), and at least one term was found for 19070 (49%) representatives from our complete data set (Supplementary Data 2). The predominant ORF annotations included *Membrane transport*, *Carbohydrate metabolism*, *Translation*, *Amino acid metabolism*, and *Replication and repair* (Supplementary Fig. 1). As expected, these functional categories were also among the top five most abundant for the MT, and top six in MP (plus *Energy metabolism*), although in a different order. The *Membrane transport* category is poorly represented in the MP (2% of the terms), which is likely explained by well-known technical issues with the gel-based sample preparation method that we used, which limits the extraction of transmembrane proteins[19]. The most abundant annotation categories mentioned above are all in line with the community function of cellulose degradation. The abundance ranking of the KO categories was assessed using the Kendall $\tau$, which takes values from $-1$ (opposite direction of the ranking) to $+1$ (total agreement in ranking). Its score is interpreted as a correlation measure; however, it is more conservative. The ranking is largely preserved from MG to MT (Kendall $\tau$: 0.77, $p < 10^{-8}$) and from MT to MP ($\tau$ 0.74, $p < 10^{-6}$) while less so from MG to MP ($\tau$ 0.68, $p < 10^{-5}$). The results show that the functional potential observed in the genomes is more preserved in the diversity of produced transcripts than in the produced proteins and thus hints to post-transcriptional regulation having an important role in addition to transcriptional regulation in prokaryotes.

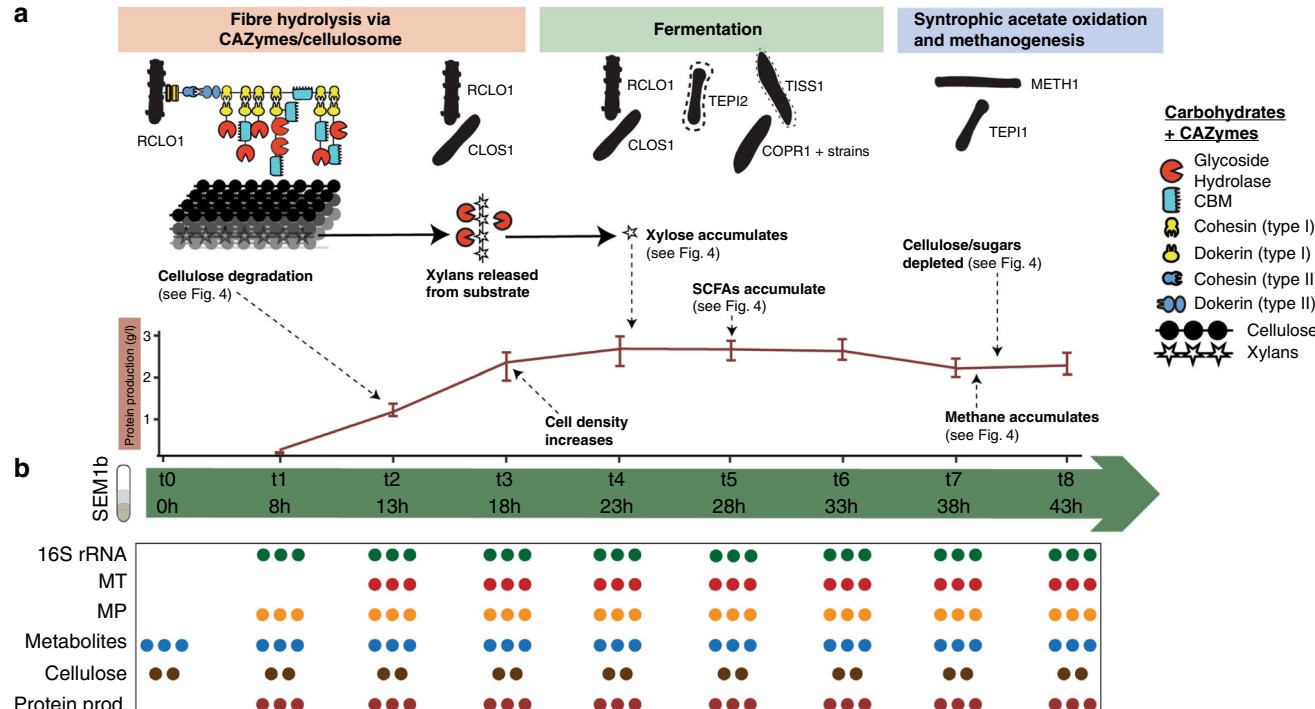

**Fig. 1 Life cycle of the SEM1b consortium and sampling scheme. a** In the SEM1b consortium, seven major microbial populations perform the metabolic processes that lead from saccharification to methanogenesis. In the first phase, RCLO1 and CLOS1 degrade the spruce substrate (predominantly cellulose and hemicellulose), thanks to a sophisticated and flexible enzymatic array, which releases simple oligosaccharides and sugars. Subsequently, the consortium grows (the protein concentration in the samples increases) up to t4 (23 h post inoculum), alongside the degradation (and fermentation) of mono- and disaccharides by RCLO1, CLOS1, TEPI1, TISS1, and COPR1+ strains. One of the sugars released from the degradation of xylan, xylose, is briefly accumulated (Fig. 4a). In addition, SCFAs are accumulated as a byproduct of microbial fermentation. In the last step, the synergetic partnership between TEP1 and METH1 (syntrophic acetate oxidizer and methanogen, respectively) converts the SCFAs and $H_2$ to methane. The bars in the protein profile represent the maxima and minima of the measurements. **b** To characterize SEM1b, 24 flasks containing spruce media were inoculated with a SEM1b culture at t0. Starting from t1 (8 h), three flasks were opened every 5 h and their content processed. From the eight time points (plus t0) different omic- and meta-data were collected (depicted in the table). Every dot represents a replicate sample, and most measurements are taken in triplicate (except for cellulose degradation).

**Protein-to-RNA ratios within a mixed-domain microbiome.** To determine whether or not microbial RNA/protein dynamics vary between ecological status (isolate vs community), metabolic state and/or taxonomic phylogeny, we quantified and resolved the numbers of transcript and protein molecules per sample (i.e., the total SEM1b consortia within each 60 ml flask, see "Methods" section), which averaged $3.8 \times 10^{12}$ (range $3.45 \times 10^{11}$–$1.10 \times 10^{13}$) and $2.2 \times 10^{15}$ (range $2.88 \times 10^{14}$–$3.46 \times 10^{15}$), respectively (Supplementary Data 3 and 4). Microbial cell volume and associated transcriptome size has been shown to change in yeast according to cell status (proliferation vs. quiescence), while the proteome is merely reshaped in its composition between these states[20]. In our case, the number of total transcripts per sample increased more than three-fold during the first 15 h (from $\sim 1.2 \times 10^{12}$ in t1 to $\sim 4.0 \times 10^{13}$ in t4) in the SEM1b consortium's life cycle and then decreased sharply, whereas the number of proteins per sample reached a plateau after 18 h post-inoculation at $\sim 2.7 \times 10^{15}$ molecules. SEM1b approximated the exponential growth phase in t3 (18 h), therefore we used the protein-to-RNA ratio from this time point for comparison against previously reported axenic estimates[2,21–24]. The replicate-averaged protein-to-RNA ratio for the bacteria in SEM1b ranges from $\sim 10^2$ to $10^4$ (median = 949, Fig. 2a), which fits the estimated range reported for *E. coli*[2]. This means that for every RNA molecule one can expect from 100 to 10000 protein molecules with a median value of 949. Our results showed a population-specific variation in the protein-to-RNA ratio within bacteria (Fig. 2a), with the median ratios for

the bacteria in SEM1b at 18h ranging from 658 in CLOS1 to 1137 in RCLO1. Although the limited number of published studies and data that enable estimation of protein-to-RNA ratios prevented our assessment of higher-resolution taxon-specific distributions within Bacteria, clear patterns were observed at a broader Domain level and are presented below (Fig. 2a).

In contrast to bacterial protein-to-RNA ratios that were relatively comparable to one another, the median protein-to-RNA ratio for an archaeal organism was ~10× higher at 12,035 protein molecules per detected RNA (Fig. 2a: METH1). The reported values for eukaryotes are 4200–5600 for yeast[21,22] and 2800–9800 for *Homo sapiens*[23,24]; therefore, we find that archaeal translation dynamics are closer to that observed within the eukaryotic domain than that of Bacteria. Structurally, the translation system of archaea more closely resembles eukaryotes[14]. Correspondingly, the RNA of Eukarya and Archaea have been shown to exhibit longer half-lives than Bacteria[25,26], with Archaea found to possess a cap complex similar to those in eukaryotes at the 5′-triphosphate end of the RNA molecule that is involved with mRNA stability[27]. Like eukaryotes, it has also been shown that archaeal RNA is regulated by post-transcriptional modification of the RNA molecule in order to upregulate protein expression[28,29]. Findings that show transcripts are present in archaeal cells for longer than bacterial cells can be used to hypothesize that this feature could have a greater role in optimizing the efficient production of protein molecules. In a microbiome setting, the greater turnover of RNA molecules and

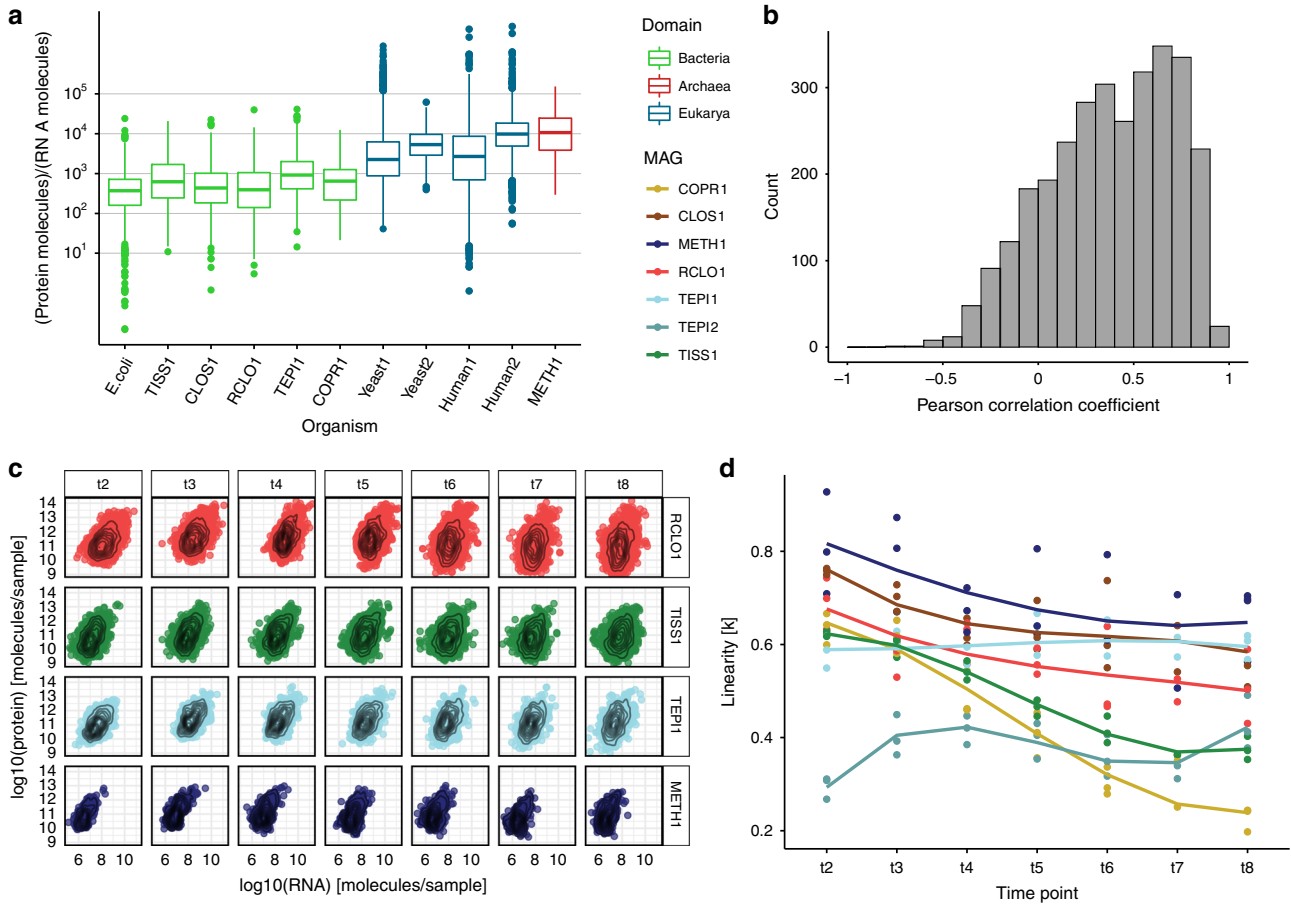

**Fig. 2 Protein-to-RNA ratio distributions within a microbial community. a** Comparison of the boxplot of protein-to-RNA ratios of selected MAGs reconstructed from the SEM1b community as well as those previously reported in the literature. (Bacteria: green, Archaea: red, Eukarya: blue). The boxes span the 25th–75th percentiles with the central bars being the medians. Whiskers extend maximum up to 1.5× the inter-quartile range or, if possible, until the most extreme of the data points. Points beyond the whiskers are considered outliers. The protein-to-RNA ratios for *E. coli* was retrieved from Taniguchi et al.[2], Yeast1 from Ghaemmaghami et al.[21], Yeast2 from Lu et al.[22], Human1 from Schwanhausser et al.[23] and Human2 from Li et al.[24]. The number of independent genes per organism used in the plot are as follows: *E. coli* = 1018, TISS1 = 587, CLOS1 = 783, RCLO1 = 799, TEPI1 = 433, COPR1 = 75, Yeast1 = 6238, Yeast2 = 6330, Human1 = 5028, Human2 = 5028, METH1 = 93. **b** The distribution of the Pearson correlation coefficients (PCC) between transcripts and their corresponding proteins computed across the time points. With a median PCC of 0.41, the change in the amount of a given transcript over time seemingly does not translate into a change in the amount of the corresponding protein. **c** Per time-point scatterplots of the absolute protein and transcript levels for ORFs that produced both detectable transcript and protein in SEM1b data sets. For simplicity, only four representative MAGs are shown, with all MAGs depicted in Supplementary Fig. 2. **d** The plot shows how the linearity parameter *k* between RNA and protein changes over time for the different MAGs. The linearity represents how a change in RNA level is reflected in a change in the corresponding protein level. The third-grade polynomial fit allows up to two bends to the curve. The parameter ranges from 0 to 1, and increasingly smaller values translate in fewer proteins being expected for the same level of RNAs. The populations CLOS1, METH1, and TEPI1 are converging toward the same values, while RCLO1 has a parallel trend, which collectively suggests the existence, and the reaching of a metabolic equilibrium among them.

lower protein–RNA ratio in bacteria could potentially facilitate their faster adaption to changes in metabolic state and substrate availabilities in their environment, at higher rates than their archaeal counterparts. However, in many complex microbiomes archaea occupy highly specialized niches such as the biological production of methane via methanogenesis, which is the energy-yielding metabolism of methanogens and is unique to the Archaea. In this context, proteins involved with hydrogenotrophic methanogenesis have been shown to be the most highly detectable in methanogens grown in co-culture with syntrophic acetate oxidizing bacteria, when compared to the same methanogen grown in axenic culture with higher concentrations of supplemented $H_2$[30]. This discrepancy between $H_2$ supply and protein levels suggests there is a requirement for methanogens to maintain highly active protein expression levels in order to keep $H_2$ at levels that are low enough to keep syntrophic acetate

oxidation energetically favorable[31]. We, therefore, speculate that methanogens, via their molecular mechanisms of maintaining high protein levels, are at an advantage to stably and efficiently maintain low $H_2$-levels, a process that is critical to the metabolic equilibrium of many microbial ecosystems[32].

In axenic culture, a microorganism is considered to be in steady state during the log phase of its growth cycle[9,33,34], specifically when the changes in proteome size are believed to be mainly dictated by a change in the transcriptome[35]. In contrary to these assumptions, comparisons of RNA and protein levels between single cells of *E. coli* grown at steady state have not been shown to correlate, however patterns do emerge when the individual cells are collectively considered at the population level[2]. In SEM1b, we wanted to see if correlations between RNA and protein levels exist across a larger microbial community and if they are affected by changes in time and life stages. We

calculated gene-wise Pearson correlation coefficients (PCCs) of protein and transcripts over time for the entire SEM1b consortium and showed that the PCC value varied greatly (Fig. 2b) with a median of 0.41. A high average $R^2$ value (0.85 for both MT and MP) was also determined between replicates indicating the stability of our results and the lack of influence from random noise. This suggested that no direct correlations between RNA and protein levels exist at any stage at a community level and that it is nearly impossible to predict the level of the given protein based on the level of the corresponding transcript.

Looking at relationships between proteome and transcriptome for individual populations within SEM1b (examples from four populations in Fig. 2c) was observed to follow a more predictable relationship, which can be described by the monomial function:

$$\text{Protein} = a \cdot \text{RNA}^k, \qquad (1)$$

The formula for log10-transformed RNA and protein levels takes the form of a linear model (see "Methods" section) that was fitted to protein and RNA distributions per time point from MAGs with the highest quality (RCLO1, CLOS1, COPR1, TISS1, TEPI1, TEPI2, and METH1) (Fig. 2d). The linearity parameter $k$ can be interpreted as the rate of which a change in RNA level is reflected in a change in the corresponding protein level. For example, if $k = 1$, a doubling in RNA level means a doubling in protein level, whereas if $k = 0.5$ a doubling in RNA level means a ~40% increase in protein level. Ranging from 0 to 1, it implies that, in the "perfect" condition where $k = 1$, the number of proteins is linked to the number of RNAs by the scalar constant $a$, whilst if $k$ approaches 0, there will be much lower expected protein levels for the same number of RNAs. With the exception of TEPI2, the linearity ($k$) between protein and RNA levels was observed to start at values above 0.5 at 13 h (t2) (Fig. 2d). The evolution of the MAGs' $k$ values over time is then divided into three groups: one where the $k$ values decrease rapidly (TISS1 and COPR1); one where they slowly decline (RCLO1, CLOS1, and METH1) and one where they stay constant if not increase (TEPI1 and TEPI2) (Fig. 2d). Notably, CLOS1, METH1, and TEPI1 are converging toward the same $k$ values, while RCLO1 has a parallel trend to them. If these trends can be used to retro-fit the steady-state definition, we can hypothesize that these four populations possess a metabolic equilibrium and that this equilibrium is approximately reached within the 10 h window between 33h and 43h (t6 and t8 respectively, Fig. 2d).

**Interplay between microbiome function and RNA/protein dynamics**. Using multi-omic data and the above-described RNA/protein dynamics, we were able to visualize that at least four populations within SEM1b converge upon a dominant metabolic state that we speculate to strongly shape the overall SEM1b community phenotype and suggest a functional co-dependence between the individual populations. To determine if this was the case, we annotated the genes and metabolic pathways for SEM1b MAGs (Fig. 3) and reconstructed their temporal expression patterns (Fig. 4). The SEM1b consortium is able to convert cellulose (and hemicellulose) to methane via the combined metabolism of its seven major constituent populations (Fig. 4a). Based on the previous analysis that showed that RCLO1 is closely related to *R. thermocellum*[10], we predict that it senses[36] its growth substrate (cellulose) and moves toward it (Fig. 4d). RCLOS1 then transcribes, translates, and secretes the components of the cellulosome, such as scaffoldins, dockerins, and carbohydrate-active enzymes (CAZymes)[37], which assemble into a dynamic multiprotein complex that degrades the substrate to smaller carbohydrates. Via the MG, we predicted that non-cellulosomal

CAZymes were also employed by the *Clostridium*-affiliated CLOS1, which acted upon the hemicellulose fraction (mainly xylan) trapped in the spruce cellulose, which was supported by the observed release of its main monomer xylose (Fig. 4a). Sugars generated via the actions of RCLO1 and CLOS1 are subsequently consumed by RCLO1, CLOS1, and *Coprothermobacter*-affiliated populations (COPR1, BWF2A, and SW3C), which were all observed to express sugar transporters (Fig. 3). Notably, CLOS1 has the most diversified transporters, making it a flexible consumer, and for the most part demonstrated highest levels of hydrolytic and fermentative gene expression after RCLO1, which again is likely tied to xylose release at later stages of the SEM1b life cycle (Fig. 4a). However, some of the transporters, such as those for oligogalacturonide, raffinose/stachyose/melibiose, and rhamnose, were not expressed, likely due to the absence of their substrates in the largely cellulose and xylan dominated spruce wood used in this study. CLOS1 was also the only population to possess the aldouronate transporter with 20 copies of gene lplA, 20 of lplB, and 16 of lplC (20/20/16) and expressing $0.4/0.7/3.8 \times 10^{10}$ and $92.8/3.5/7.0/ \times 10^{11}$ combined median transcripts and proteins per sample; making it one of the few transporters detectable at the protein level. Similarly, the *C. proteolyticus* strains (BWF2A and SW3C) possess and express unique sugar transporters, likely gaining access to an undisputed pool of arabinogalactan or maltooligosaccharide. The transporter for pentamers ribose/xylose was the most common and possessed by RCLO1, *C. proteolyticus* populations and *Tepidanaerobacter* populations (TEPI1 and TEPI2). Notably from Fig. 3, it is clear that the proteins from the transporters are almost never found in the samples, even if the respective RNAs are present in the data set. This is likely due to the difficulties in extracting transmembrane proteins[19] with the gel-based method that we employed.

The process of degrading cellulose and simple saccharides via hydrolysis and fermentation ultimately results in the production of short-chain fatty acids (SCFAs) such as propionate, butyrate, and acetate, which are subsequently metabolized by the predicted SCFA-oxidizing populations in SEM1b (TISS1, TEPI1, TEPI2) (Fig. 4a). The only metabolically active SCFA-oxidizing population in SEM1b was predicted to be TEPI1, as it demonstrated high $k$ values that increased over time (Fig. 2d) and harbored a complete Wood–Ljungdahl pathway that was detectable in both MT and MP (Fig. 3). It has been shown that oxidizers can improve the oxidation of SCFAs (up to double speed) when superior NADPH and ATP generators (e.g., glucose) are consumed in small amounts to complement the stoichiometry through the pentose phosphate pathway (PPP) without triggering the shift of the entire cell's metabolism toward another substrate[38]. In this context, it is interesting to note that TEPI1 was the only MAG that encoded and expressed a hexose (allose) transporter (Fig. 3). Aldohexoses (such as D-allose, D-glucose, D-mannose, etc.) are imported and transformed into fructose-6P in two reactions (both expressed in TEPI1), which can then be fed into either the PPP or the Glycolysis pathways. Xylose is a product of the degradation of hemicellulose present in our system (Fig. 4a) and can be converted to ribulose-5P and fed to the PPP in three reactions. This data, in combination with a highly expressed and detectable Wood–Ljungdahl pathway over time (Fig. 4a), points to the establishment of TEPI1 as the only syntrophic acetate oxidizing bacteria in the SEM1b consortium. We speculate that TEPI1's syntrophic metabolism is helped by the other SEM1b populations that generate acetate as a fermentation end-product and the supply of sugars released by the cellulosomal complex such as glucose and xylose. Interestingly the closely related MAG TEPI2 was observed to lack the Wood–Ljungdahl pathway and to express ~10 times more transcripts for the ribose/xylose transporter than TEPI1;

relegating it to the role of mere sugar degrader, and probably scavenger in the community.

Although TISS1 seems mostly to phase out of the community and its $k$ value associated with its protein to transcript

relationship (Fig. 2d), TEPI2 implements an exit strategy in the form of sporulation. All the Gram-positive populations from the SEM1b consortium (RCLO1, CLOS1, TISS1, TEPI1, and TEPI2) were able to produce spores and express the spore marker *spoIV*,

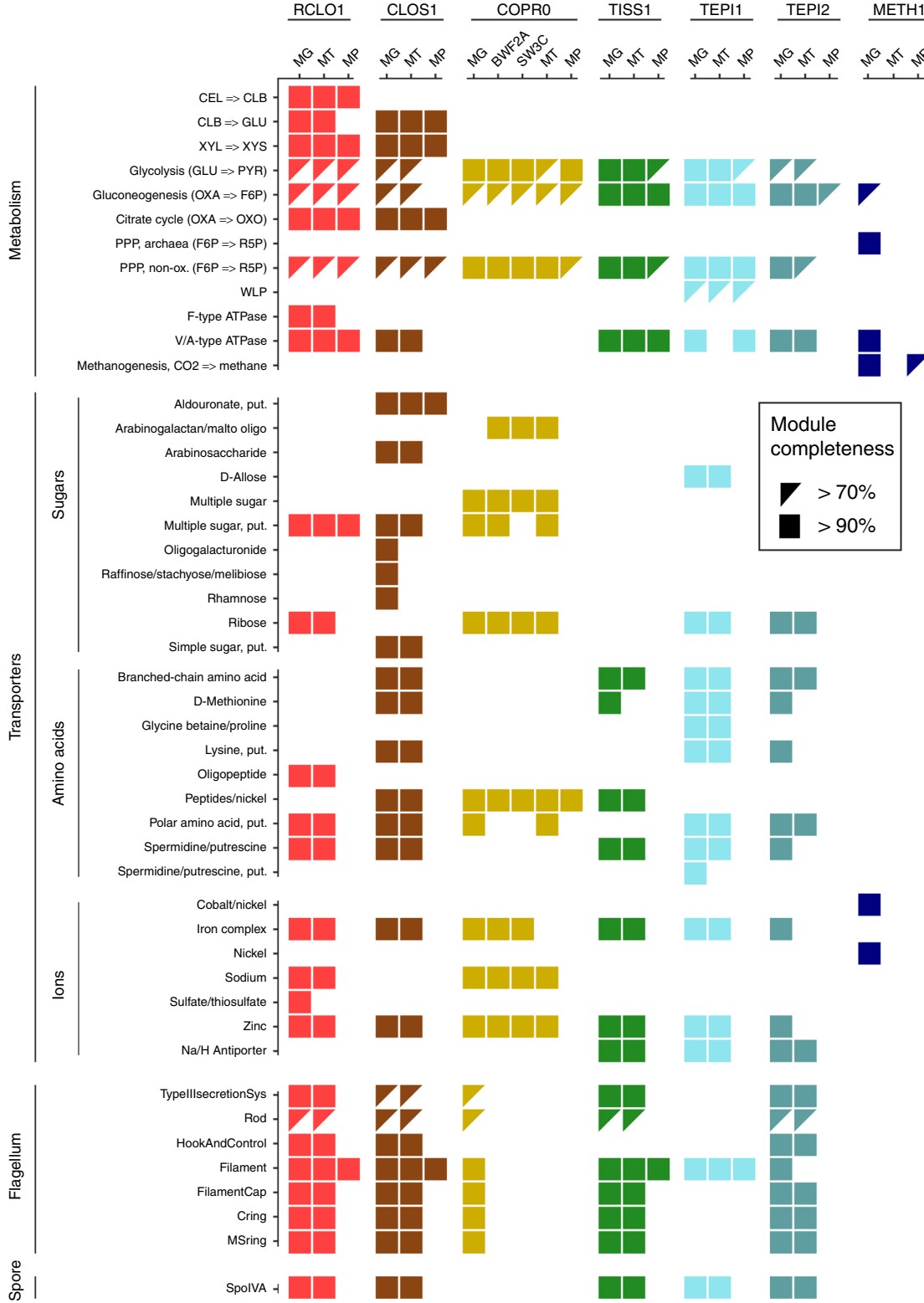

**Fig. 3 Overview of the genetic potential and expressed modules in SEM1b.** Module completeness denotes the level of detected RNA and proteins mapped to major genes/metabolic pathways that are critical to the SEM1b life cycle. Triangles denote completeness greater than 70% while square completeness >90%. Only MAGs with the highest quality reconstruction (RCLO1, CLOS1, COPR1, TISS1, TEPI1, TEPI2, and METH1) are included as well as two isolated and genome-sequenced *Coprothermobacter* strains, for which the transcriptome and the proteome were considered as the species level.

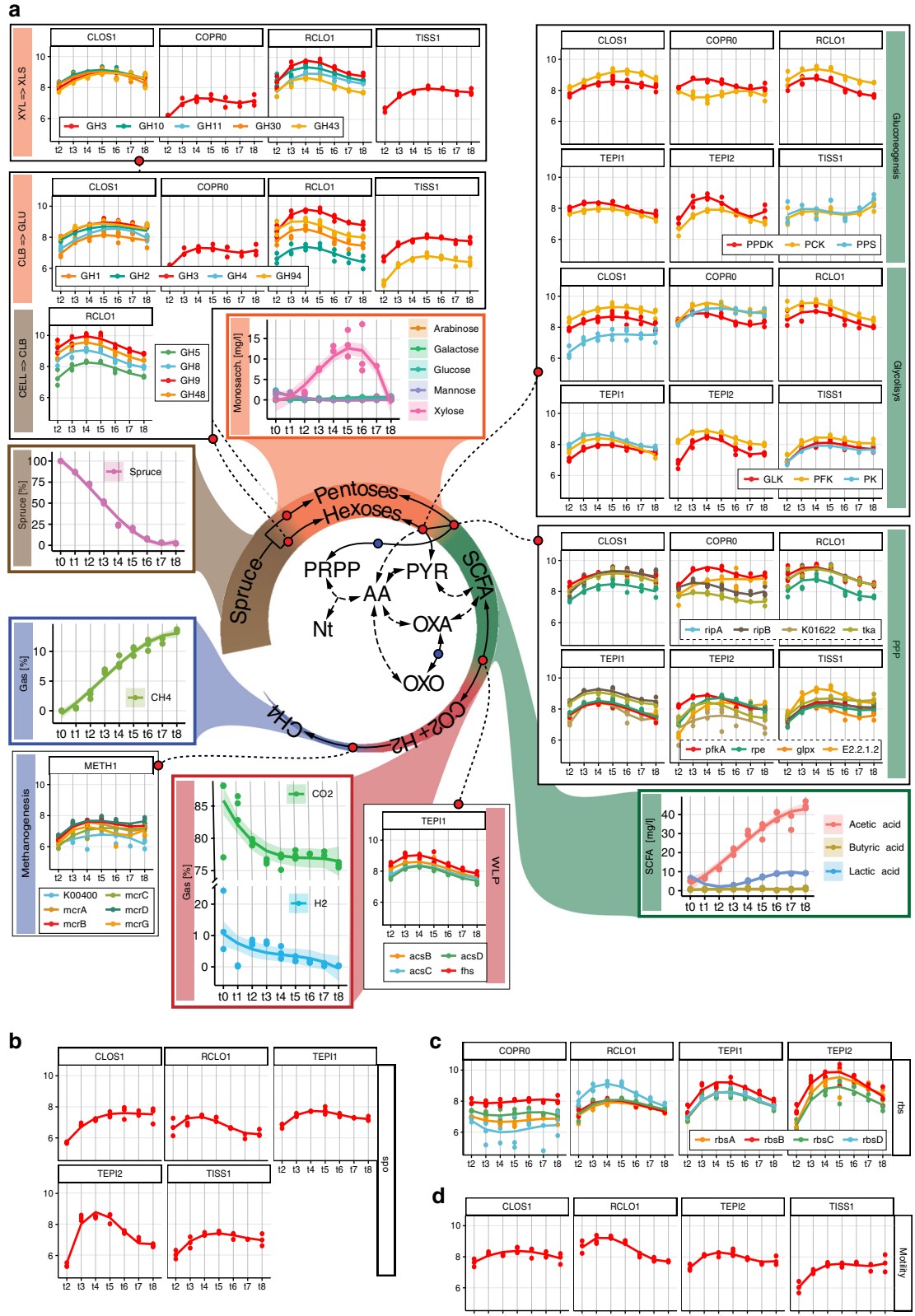

an ATPase associated to the surface of the neospore that promotes the assembly of the coating and is common to all the spore-forming bacteria[39] (Fig. 4b). TEPI2 however increased the level of transcripts for spoIV by 1000 times within the 13 h and the 18 h time points, reaching the maximum at 23 h, and having a production ten times higher than the phylogenetically related

TEPI1. All SEM1b populations, except the *C. proteolyticus* isolates and TEPI1, have the genetic potential for flagellar synthesis but the respective transcripts were only observed for RCLO1, CLOS1, TISS1, and TEPI2. The filament protein of RCLO1 is by far the most abundant protein in the samples with an average of $2.8 \times 10^{13}$ molecules per sample, which matches the

**Fig. 4 Schematic representation of the temporal and co-dependent metabolism of SEM1b. a** Within SEM1b, four major metabolic stages are required: Spruce → Hexoses/Pentoses, Hexoses/Pentoses → SCFAs, SCFAs → $CO_2 + H_2$, and $CO_2 + H_2$ → methane. Metabolites (spruce, sugars, SCFAs, $CO_2 +$ $H_2$, and methane) involved in these processes were measured and the temporal analysis of the metabolic pathways involved in their interconversion is depicted for the major SEM1b populations. Other metabolites (for which abbreviations are: Nt = Nucleotides, PRPP = Phosphoribosyl pyrophosphate, AA = Amino acids, PYR = Pyruvate, OXA = Oxaloacetate, and OXO = Oxo-glutarate) are shown to highlight the essential metabolism of the microbes. In the central metabolic network, the metabolites are linked by solid arrows if the interconversion requires one step or the link between them is addressed more in detail (blue dot if in Fig. 3, red dot if in a pathway plot herein). Metabolic pathways are quantified via marker genes (selection in "Methods" section) in the scale of log10-transformed transcript molecules per sample (values depicted in the y axis) while the solid lines in the plots represent the cubic fitting of the data points. The shaded area on the metabolites' plots represents the 95% confidence interval of the curve fitting. More metabolites' abbreviations are CELL = Cellulose, CLB = Cellobiose, GLU = Glucose, XYL = Xylan, XLB = Xylobiose, and pathways' abbreviations are WLP = "Wood–Ljungdahl pathway", PPP = "Pentose Phosphate pathway". **b** Sporulation is common to all Gram-positive bacteria in the community and it is quantified with the marker *spoIVA*. Notably, TEPI2 is investing greatly in spore formation until 28 h after the inoculum (t4). **c** The genes for the ribose and xylose transporter (*rbs*) are expressed in four populations. Notably, TEPI2 produces more rbs transcripts than the closely related MAG TEPI1; indeed, the former has been predicted to be a mere fermenter whilst the latter bases its metabolism on the Wood–Ljungdahl pathway (Fig. 4a). **d** Microbial motility is represented by the marker gene *flgD*. RCLO1 is the most active bacterium, producing less flagella over time after t4. For **b–d**, RNA expression uses the same scale as **a**.

idea of RCLO1 investing in motility to reach the cellulose fibers and starting with the highest level of marker flgD in the community (Fig. 4d).

In microbial ecosystems, acetate is oxidized by secondary fermenters to $CO_2/H_2$ or formate, a process that is mediated by the Wood–Ljungdahl pathway in reverse. The oxidation of acetate associated with the reverse WLP is coupled with the transition between NADH/NAD⁺, and translocates Na⁺ to create an electrochemical gradient, which is then used by the type-V ATPase to synthesize ATP[40]. Indeed the NAD⁺-Fd$_{red}$-dependent Na⁺ translocation system *rnf* is expressed in both the fermenting and syntrophic acetate oxidizing bacteria of SEM1b, while type-V ATPase, which produces energy by exploiting the Na⁺/H⁺ gradient, were detected in all the SEM1b populations aside from METH1 and *C. proteolyticus*-affiliated populations (Fig. 3). Moreover, the TEPI1 MAG expresses the NAD⁺ (NADP⁺)-reducing hydrogenases complex, which reduces hydrogen ions to $H_2$ using NAD(P)H as the electron donor. The molecular hydrogen generated here would then be used by the syntrophic partner METH1 to form methane (Fig. 4a). However, acetate oxidation that is mediated by the reverse Wood–Ljungdahl pathway is thermodynamically unfavorable unless coupled with syntrophic hydrogenotrophs. Within SEM1b, the METH1 population is a hydrogenotrophic methanogen and the methanogenesis pathway, which is observed in the METH1 MG and MP, is the largest pathway in SEM1b according to the number of genes involved ($n = 112$). In METH1, we also observed transporters for nickel, the metal ion found in the F$_{430}$ prosthetic group in the methyl-coenzyme M reductase complex (McrABG), which is responsible for the terminal step in anaerobic methanogenesis[41]. Transporters for another key metal, cobalt, which is utilized by cobalamin-requiring enzymes such as the energy-conserving methyl-H$_4$MPT:CoM-SH methyltransferase complex (MtrABCDEFGH), were also detected in the MG and MP of METH1. Within hydrogenotrophic methanogens, electrochemical gradients generated Na⁺ ion exclusion by the Mtr complex allows for the inflow of Na⁺ through ATP synthases to generate energy. Surprisingly, no H⁺/Na⁺ *nha* ion transporter, which is commonly observed in methanogens were observed in the METH1 MG, MT, or MP. The Na⁺/H⁺ antiporter *nha* was encoded and expressed in populations TEPI1, TEPI2 and TISS1 (Fig. 2), which does point to an important role of these ions in the bacterial component of the SEM1b consortium. Overall, our more classical pathway-wise exploration of the SEM1b populations supported that RCLO1, CLOS1, TEPI1, and METH1 indeed share functional co-dependencies and supported our predictions via RNA/protein dynamics that they converge upon a dominant metabolic state.

**Translation control impacts cell status and resource usage**. A change in protein regulation can be causally linked to a change in the population status (steady state or transition). Within the cell, proteins are predominately the performers of cellular functions thus the change in cell status can be achieved by actively altering the protein level. In addition to protein-to-RNA ratio assessments, our absolute multi-omics analysis allowed us to explore the crucial aspect of protein-level regulation, which is poorly understood in microbiomes. The control of protein levels in bacteria is believed to occur via transcription control, "control by dilution"[42] (dispersal of proteins via subsequent cell divisions), sRNA activity[43], and rarely by protein degradation[44]. Similar to transcription control, translation can also be controlled by a dynamic pool of translational factors, such as initiation, elongation, and ribosome components[45]. The processes targeted by these systems require a rapid change in the number of proteins in the cell that cannot wait for a change in RNA levels or a dilution effect. We used our absolute quantifications of SEM1b transcripts and proteins as well as PECA-R[46] to predict the "change-point", which takes into account estimates of protein translation and degradation rates (Supplementary Data 5). The analysis found 305 significant rate changes, accounting for 302 ORFs. Of the rate changes', 94% were downregulated and 71% of the ORF were functionally annotated. RCLO1 has 28 downregulated ORFs between 13 and 18 h (t2–t3), mostly from complexes involved in chemotaxis (*cheY*, *cheW*, *mcp*), flagellum assembly (*flgG*, *flgK*, *fliD*) and shape determination (*mreB*). In the following 5 h several systems concerning carbon fixation are affected, such as phosphoglycerate kinase (PGK), triosephosphate isomerase (TPI), phosphate acetyltransferase (EC 2.3.1.8), isocitrate dehydrogenase (IDH1), and pyruvate orthophosphate dikinase (PPDK). In the next 5 h, it downregulates the translation of the cell division protein ZapA as well. The reduction in protein production for chemotaxis, mobility, and then cell division matches the idea that within 13 h of the inoculation, RCLO1 sensed, reached, and colonized the cellulose fibers. Contextually the release of medium length oligosaccharides enables RCLO1 to engage in the more energetically favorable fermentation metabolism. TISS1 has a decrease in translation rates of ORFs related to metabolic processes between 13 and 18 h, mostly involving cofactors (*fhs*, *folC*, *folD*, *lplA*, *metH*, *pdu0*, and *nadE*) and amino acids (*aorQ*, *hutI*, LDH, *metH*, *mtaD*, and *pip*). TEPI1 downregulated 60 ORFs, accounting for part of its carbohydrate metabolism (e.g., PGK, TPI), the amino acid transporters, and the NADH dehydrogenase complex (NDH). TEPI2 has 19 ORFs subject to downregulation in the 13–18h interval, such as pyruvate ferredoxin oxidoreductase (PFOR), GK, fructose-bisphosphate aldolase (FBA), tansaldolase EC 2.2.1.2, and the ribose/xylose transporter subunit

*rbsB*. In the last interval (33–38 h), RCLO1 upregulated the translation of 10 ORFs, among which the flagellar FlbD and shape determination (*mreB*); seemingly starting to restore the functions downregulated in the 13h-18h interval.

## Discussion

We present the reconstruction of microbiome from a model environment and quantified the number of RNAs and proteins per sample over time in absolute terms and as ratios. The observed discrepancy between RNA and protein levels of a given gene within the SEM1b consortium (Fig. 2b) could raise questions as to which omic technology (transcriptomics or proteomics) should be applied to assess community function. We would argue that both technologies have merit and if possible, should be applied to the same sample(s), given that transcript levels store the "recent history" (up to minutes) of a cell and/or the community at large, whilst proteins usually remain viable much longer (up to hours) and can result in a different interpretation of function. In addition, relative quantification of omic data is much more commonly used and reported in microbiology studies and would have largely revealed the same changes in expression patterns that were highlighted in Fig. 4. However, our absolute approach enabled us to assess and report the protein-to-RNA ratio of multiple microbial populations simultaneously, which individually engage in distinct, yet integrative metabolic pathways that ultimately cumulate into the community's principal phenotype of converting cellulose to methane. We extended the results from Taniguchi et al.[2], showing that our populations had a varying protein-to-RNA ratio in the predicted interval of $10^2$–$10^4$ while presenting for the same quantity for an archaeal population (METH1): $10^3$–$10^5$, which resembled the previously measured values for eukaryotes[21–24]. The greater ecological significance of the seeming archaeal capacity to generate higher protein levels at a lower "RNA-cost" is of interest, as many archaeal populations in mixed-domain microbiomes are known to occupy essential ecological niches and exert strong functional influence, despite their cell concentrations being orders of magnitude lower than their bacterial counterparts (i.e., methanogens in the rumen microbiome[47]).

In addition, we assessed the *k* value (proxy for linearity) associated to transcriptome and proteome for each population over time (Eq. 1), finding that three major populations of the community, a fermenter (CLOS1), a syntrophic acetate oxidizing bacterium (TEPI1) and a methanogen (METH1), were converging on the same values in parallel with the primary cellulose degrader (RCLO1) (Fig. 2d). The highlight of their seemingly intertwined RNA/protein dynamics matches with their metabolic complementarity, starting from RCLO1 degrading cellulose to sugars and SCFAs, CLOS1 fermenting sugars to SCFA, TEPI1 oxidizing SCFAs to $H_2$, and METH1 converting $CO_2$ and $H_2$ to methane. Closer examination revealed even more intricate relationships involving $Na^+$ and $H^+$ ions as well as secondary sugars (i.e., xylose) reiterating that each population needs the metabolic activity and subsequent byproducts of the previous one to provide a supply of growing metabolites (Fig. 4a). Moreover, the estimation of translation and/or protein degradation rates pointed at a translational negative control for several ORFs involved in chemotaxis/motility and central metabolism, marking important changes in the community status. In conclusion, our data highlights that simple modifications to multi-omics toolkits can reveal much deeper functional-related trends and integrative co-dependent metabolisms that drive the overall phenotype of microbial communities, with the potential to be expanded to more-complex and less-characterized microbial ecosystems.

## Methods

**Background and multi-omics sampling**. A microbial consortium called SEM1b was obtained from a biogas reactor using serial dilution and enrichment methods on spruce cellulose. Metagenomic analysis was initially performed on the SEM1b community using two different generations that had consistent population structure and was used as a subsequent SEM1b time-series experiment. The time series analyses consisted of metabolomics, metaproteomics, and metatranscriptomics over nine time points (at t0, 8, 13, 18, 23, 28, 33, 38, and 43 h) in triplicate (A, B, and C), spanning the consortium life cycle (Fig. 1b). For every time point (60 ml), a 6 and 30 ml aliquot was collected and used for MT and MP analysis, respectively. The RNA internal standard was added to the 6 ml aliquot and the resulting transcript levels were therefore multiplied by 10 to reconstruct the original 60 ml sample size (3× replicates). In the case of the MP analysis, the proteins were extracted from the 30 ml aliquot, and the protein concentration calculated using the Bradford method from which we computed the original mass of protein in the 60 ml sample (3× replicates). Therefore, the number of transcripts and proteins used in the paper refers to the whole consortium contained within each culture flask.

**Metagenomics data acquisition**. For the generation of metagenomic data, 6 ml samples of SEM1b culture were taken and cells were pelleted prior to storage at −20 °C. A cell pellet was produced by centrifugation of 2 ml of samples at 14,000×*g* for 5 min. Pellet was resuspended in 1 ml of RBB + C buffer (500 mM NaCl, 50 mM Tris–HCl; 50 mM EDTA, 4% SDS) and incubated for 20 min at 70 °C. NaCl solution was used to reach 0.7 M and 1:10 volume of CTAB buffer was added before an additional incubation at 70 °C for 10 min. An equal volume of Chloroform is then added and centrifuged at 14,000 rcf for 15 min. The aqueous phase was retrieved and an equal volume of phenol:chloroform:isoamylalcohol (25:24:1) was added and centrifuged at 14,000 rcf for 15 min. The aqueous phase was retrieved one more time and 2 volumes of 95% ethanol were added and gently mixed until the DNA spooled and could be transferred with a sterile loop to a tube containing 200 μl of 70% ethanol. After centrifugation at 14,000 rcf for 2 min, the supernatant was discarded, and the pellet air-dried prior to being resuspended into 30 μl of TE buffer (pH 8.0). DNA samples were prepared with the TrueSeq DNA PCR-free preparation and sequenced with paired-ends (2 × 125 bp) on one lane of an Illumina HiSeq3000 platform (Illumina Inc) at the Norwegian Sequencing Center (NSC, Oslo, Norway). The reads were 3′-trimmed (Phred < 20, length > 100) with cutadapt[48] and filtered using FASTX-Toolkit (http://hannonlab.cshl.edu/fastx_toolkit/) to retain the reads with Phred > 30 on at least 90% of their length. The reads were mapped (ID = 100%) on two *Coprothermobacter proteolyticus* isolates from SEM1b using the Burrows–Wheeler Aligner with maximal exact matches (BWA-MEM)[49]. The remaining reads were assembled with MetaSpades v 3.10.0 (*k*-mers: 21, 33, 55, 77)[50] and the contigs binned with Metabat v0.26.3 (in "very sensitive mode"). The contigs were also uploaded to the Microbial Genomes and Microbiomes[51] system for gene prediction and annotation. Resulting annotated open reading frames (ORFs) were retrieved and used as a reference database for the metatranscriptomic and metaproteomic analysis.

**Metatranscriptomics data acquisition**. mRNA extraction was performed in triplicate on time points t2–t8. The extraction of the mRNA included the addition of an in vitro transcribed RNA as an internal standard to estimate the number of transcripts in the natural sample compared with the number of transcripts sequenced. For further normalization, total RNA was extracted using enzymatic lysis and mechanical disruption of the cells and purified with the RNeasy mini kit (Protocol 2, Qiagen, USA). The RNA standard (25 ng) was added at the beginning of the extraction in every sample. After purification, residual DNA was removed using the Turbo DNA Free kit following the manufacturer's instructions. Free nucleotides and small RNAs such as tRNAs were cleaned off with a lithium chloride precipitation solution according to ThermoFisher Scientific's recommendations (https://www.thermofisher.com/be/en/home/references/ambion-tech-support/rna-isolation/general-articles/the-use-of-licl-precipitation-for-rna-purification.html) Briefly, one volume of cold 5 M LiCl solution was added to the samples, mixed well, and incubated at −20 °C for 30 min. Samples were centrifuged at maximum speed for 30 min at 4 °C. The supernatants were discarded and the pellets were washed with 70% ethanol prior to being resuspended in 16 μl of RNase-free water. To reduce the number of rRNAs, the samples were treated to enrich for mRNAs using the MICROBExpress kit (Applied Biosystems, USA). The successful rRNA depletion was confirmed by analyzing both pre- and post-treated samples on a 2100 bioanalyzer instrument. The enriched mRNA was amplified with the MessageAmp II-Bacteria Kit (Applied Biosystems, USA) following the manufacturer's instruction and sent for sequencing at the NSC (Oslo, Norway). Samples were subjected to the TruSeq stranded RNA sample preparation, which included the production of a cDNA library, and sequenced with paired-end technology (2 × 125 bp) on one lane of a HiSeq3000 system.

The resulting sequences were checked for overrepresented features with FastQC (www.bioinformatics.babraham.ac.uk/projects/fastqc/); features and low qualities (Phred < 20) ends were trimmed using Trimmomatic[52] v.0.36. The reads were then filtered using an average Phred >20 in a 10nt window and a minimum length of 100 nt. The rRNA and tRNA reads were removed using SortMeRNA[53] v2.1b. Also, the reads mapping on the internal standard pGEM-3Z were extracted using

SortMeRNA, and their counts used as $I_R$ in the "Functional omics absolute quantification" section of the "Methods", while the reads that did not map (i.e., the transcriptome in the sample) were used as $\sum T_R$. The retained reads were mapped against the predicted genes-data set using Kallisto pseudo –pseudobam[54] and the mapping files were produced with bam2hits. Transcripts were quantified with mmseq[55] and collapsed using mmcollapse[56]. The collapse first gathers ORFs into groups if they have 100% sequence identity, and in a second round the ORFs (already termed ORFGs) are collapsed if they acquire unique hits as a group. A summary of the filtering and mapping output is summarized in Supplementary Data 6 and 7, respectively.

**Metaproteomics data acquisition.** Proteins were extracted from t1 to t8 in triplicate. From each sample, 30 ml of culture containing cells and substrate was centrifuged at $500 \times g$ for 5 min to pellet the substrate. The supernatant was centrifuged at $9000 \times g$ for 15 min to collect the cells. Cell lysis was performed by resuspending the cells in 1 ml lysis buffer (50 mM Tris–HCl, 0.1% (v/v) Triton X-100, 200 mM NaCl, 1 mM DTT, 2 mM EDTA) and keeping them on ice for 30 min. Cells were disrupted in $3 \times 60$ s cycles using a FastPrep24 (MP Biomedicals, USA) with glass beads (size, ≤106 μm). Debris were removed by centrifugation at $16,000 \times g$ for 15 min. The supernatants containing the proteins were kept at −20 °C until further processing. Extracted proteins were quantified using Bradford's method (in triplicate), which quantified the samples by combining 2–10 μl of protein extract with 20 mM Tris HCl (pH 7.5) to reach 800 μl, with 200 μl of BioRad Essay solution subsequently added. Samples were vortexed, centrifuged briefly, and let to rest for 5 min before measuring with dedicated cuvettes. Blanks composed of 800 μl of buffer and 200 μl of BioRad Essay solution were used before each set of measurements. 50 μg of each sample were denatured using SDS sample buffer and loaded on an Any-kD Mini-PROTEAN gel (BioRad Laboratories, USA) and separated by SDS-PAGE for 20 min at 270 V. Each gel lane was cut into 16 slices and the reduction, alkylation, and tryptic digestion of the proteins into peptides were performed in-gel. It is important to note that previous reports have shown that absolute protein quantification may be biased by the protease selected for digestion[57]. While ideally, more than one protease could have been used, we have used trypsin in our analysis (which is commonplace in most proteomics experiments) and obtained a high correlation of absolute protein quantification between replicates (average $R^2 = 0.85$, with the outlier t7C removed). Moreover, the protein-to-RNA ratios observed for the different bacteria and archaea correlate well with previous literature (E. coli, yeast, human), indicating that our absolute quantifications are on par. The tryptic peptides were extracted from the gel and desalted prior to mass spectrometry analysis. Peptides were analyzed using a nanoLC-MS/MS system consisting of a Dionex Ultimate 3000 UHPLC (ThermoScientific, Germany) connected to a Q-Exactive hybrid quadrupole-orbitrap mass spectrometer (ThermoScientific, Germany) equipped with a nanoelectrospray ion source. The samples were loaded onto a trap column (Acclaim PepMap100, C18, 5 μm, 100 Å, 300 μm i.d. ×5 mm, ThermoScientific) and backflushed onto a 50-cm analytical column (Acclaim PepMap RSLC C18, 2 μm, 100 Å, 75 μm ID, ThermoScientific). At the start, the columns were in 96% solution A [0.1% (v/v) formic acid], 4% solution B [80% (v/v) acetonitrile, 0.1% (v/v) formic acid]. The peptides were eluted using a 90 min gradient developing from 4 to 13% (v/v) solution B in 2 min, 13 to 45% (v/v) B in 70 min, and finally to 55% B in 5 min before the wash phase at 90% B, all at a flow rate of 300 nl/min. The Q-Exactive mass spectrometer was operated in data-dependent mode and the 10 most intense peptide precursors ions were selected for fragmentation and MS/MS acquisition. The selected precursor ions were then excluded for repeated fragmentation for 20 s. The resolution was set to $R = 70,000$ and $R = 35,000$ for MS and MS/MS, respectively.

A total of 384 raw MS files (8 samples × 3 biological replicates × 16 fractions) were analyzed using MaxQuant[58] version 1.4.1.2 and proteins were identified and quantified using the MaxLFQ algorithm[59]. The data were searched against the generated MG data set supplemented with common contaminants such as human keratin and bovine serum albumin. In addition, reversed sequences of all protein entries were concatenated to the database for the estimation of false discovery rates. The tolerance levels for matching to the database was 6 ppm for MS and 20 ppm for MS/MS. Trypsin was used as a digestion enzyme, and two missed cleavages were allowed. Carbamidomethylation of cysteine residues was set as a fixed modification and protein N-terminal acetylation, oxidation of methionines, deamidation of asparagines and glutamines, and formation of pyro-glutamic acid at N-terminal glutamines were allowed as variable modifications. The "match between runs" feature of MaxQuant[59] was applied. When proteins cannot be unambiguously identified with unique peptides, MaxQuant will group them and quantify them together as one ORFG. All identifications were filtered in order to achieve a protein false discovery rate (FDR) of 1%. Quantitative information was retrieved using the LFQ intensities of each protein.

**Metabolomics data acquisition.** For monosaccharide detection, 2 ml samples were taken in triplicates, filtered, and sterilized with 0.2 μm sterile filters, and 15 min boiling. Soluble sugars were identified and quantified by high-performance anion-exchange chromatography (HPAEC) with pulsed amperiometric detection

(PAD). For quantification, peaks were compared to linear standard curves generated with known concentrations of selected monosaccharides (glucose, xylose, mannose, arabinose, and galactose) in the range of 0.001–0.1 g/l.

For the short-chain fatty acids (SCFAs), 1 ml was taken in triplicate from each time point, they were centrifuged at $16,000 \times g$ for 5 min and the supernatants were filtered with 0.2 μm sterile filters. 5 μl of sulfuric acid 72% were added to the filtrates and let at rest for 2 min before being centrifuged again at $16,000 \times g$ for 5 min, transferred in a new tube, and stored at −20 °C until processing. SCFAs were then measured at 210 nm using a Dionex 3000 HPLC with a Zorbax Eclipse Plus C18 column from Agilent Technologies (150 × 2.1 mm (3.5 mm particles)) and operated at 40 °C. The VFAs were eluted isocratically with 100% methanol and 2.5 mM $H_2SO_4$ at 0.3 ml/min flow rate.

**Absolute metatranscriptomics quantification.** The absolute quantification of transcripts was taken from Mortazavi et al.[11] using the internal standard from Gifford et al.[12] as a reference to estimate the length of the initial transcriptome. The number of reads produced in a given sample is proportional to the total amount (in Nt) of starting material. With the addition of an internal standard we have the following proportion between the starting material for transcripts expressed in nucleotide length ($T_{Nt}$) and the internal standard ($I_{Nt}$) and the reads they produce ($T_R$ and $I_R$, respectively):

$$\frac{\sum T_{Nt}}{\sum T_R} = \frac{\sum I_{Nt}}{I_R},$$

in which the sums are taken over a single sample. The formula can be rearranged as:

$$\sum T_{Nt} = \sum I_{Nt} \times \frac{\sum T_R}{I_R}.$$

Since we know the number of molecules of internal standard added ($I_M$) and its length ($I_{Nt}$), we can substitute them in the equation as:

$$\sum T_{Nt} = I_M \times I_{Nt} \times \frac{\sum T_R}{I_R}.$$

We can now use the estimation of the starting length of the transcriptome and the RPMK transcript measurements in the formula from Mortazavi et al.[11]:

$$T_M = \frac{T_{RPMK}}{10^9} \times \sum T_{Nt},$$

which becomes:

$$T_M = \frac{T_{RPMK}}{10^9} \times I_M \times I_{Nt} \times \frac{\sum T_R}{I_R}.$$

**Absolute metaproteomics quantification.** The "Total protein approach" method from Wiśniewski et al.[13] relies on the use of the protein mass per sample, the computed molecular weight (MW) of the detected proteins to transform the LFQ values into absolute ones. Here we omitted the per-cell quantification since SEM1b is a heterogeneous community and MG measurements were not taken for the time series.

We computed the Total protein$_i$ as:

$$\text{Total protein}_i = \frac{\text{LFQ intensity}_i}{\sum \text{LFQ intensity}}.$$

Then the Protein concentration$_i$ was obtained from the previous with:

$$\text{Protein concentration}_i = \frac{\text{Total protein}_i}{\text{MW}_i}.$$

The method was developed on the assumption that the reference proteome is complete and that the total mass of the peptides detected is equal to the total mass of peptides processed by the machine. This is not necessarily valid in a microbiome for which the reference cannot be completely reliable. Knowing the Total protein mass [g] per sample, computed from the protein concentration estimated with the Bradford assay and the starting volume of the culture (60 ml), we estimated the Detected protein mass [g] (i.e., how much of protein mass is explained by the peaks recognized during MP analysis) using the raw MP files with the following formula:

$$\text{Detected protein mass} = \frac{\text{Total protein mass} \times \sum_{i=1}^{\text{Pep}_{\text{id}}} \text{Base peak intensity}_i \times \text{Mass}_i}{\sum_{j=1}^{\text{Pep}_{\text{tot}}} \text{Base peak intensity}_j \times \text{Mass}_j},$$

where the Base peak intensity$_i$ and Mass$_i$ correspond to the homologous values for the $i$th identified peak (i.e., a peak with an amino acid sequence associated, thus it can be called a peptide) in the raw MP files.

Finally, the copy number of proteins per sample was computed using the Avogadro's number ($N_A$) as:

$$\text{Copy number}_i = \text{Protein concentration}_i \times \text{Detected protein mass} \times N_A.$$

**Multi-omics data set integration.** The MT and MP data sets estimate the absolute abundance of ORFGs over time. An expression group is defined in this study as a set of ORFs, which cannot be further resolved using the available data. When the

analysis required the direct comparison of ORFs (e.g., transcript–protein correlation and protein-to-RNA ratios) only the singleton subset of the ORFGs was considered. The subset may suffer marginally from a loss in data points (ORFs), however the genomes in which this phenomenon had a larger impact (COPR2-3, BWF2A, and SW3C) were not used to estimate numerical properties such as protein-to-RNA ratios and $k$ values. In addition, the impact of data loss for the aforementioned MAGs/strains was illustrated in Supplementary Fig. 2 and did not outline any clear distribution that was opposing the observations made for the MAGs used in this study. The reliability of the expression estimation is linked to the number of unique hits (reads or peptides) available for a given ORF, therefore all the entries with 0 unique hits were filtered out. The data sets were then log10-transformed with a pseudocount equal to one. After expression density plotting, the minimum expression thresholds of 5 and 8 were selected for MT and MP, respectively, and the data were filtered accordingly. The principal component analysis was used to screen the samples and t7C (time point 7, replicate C) was identified as an outlier and removed before downstream analysis.

**MP/MT linear fit**. We took the intersection of ORFs present in the MT and MP layers of the data set for each of the selected MAGs (COPR1, CLOS1, COPR1, METH1, RCLO1, TEPI1, TEPI2, TISS1), and, for each sample, we performed a regression analysis in R. The values span several orders of magnitudes, thus we decided to fit the monomial functional:

$$\text{Protein} = a \cdot \text{RNA}^k,$$

which can be rewritten as:

$$\text{Log(protein)} = a + k \cdot \log(\text{RNA}),$$

to be more easily fitted as a linear model. The previously log10-transformed protein levels were used as $y$ while the log10-transformed RNA was used as $x$ in a linear model using the lm function. The slopes of the models were then used to fit a third-grade polynomial function to obtain the $k$ value change profile in Fig. 2d.

**Functional annotation and module completeness**. The KEGG Orthology (KO) numbers were assigned to the ORFs as a part of the annotation pipeline from IMG[51]. The ORF-wise annotation was then translated into the MT/MP-ORFGs assigning to each ORFG a non-redundant set of all the terms assigned to all the ORFs in the group. We used the KO numbers to estimate the KEGG module completeness using the R package MetaQy[60] v.1.1.0. The Glycosyl Hydrolases annotation was retrieved from Kunath et al.[10].

**Metabolic marker genes selection**. The metabolic marker genes for Fig. 2 were selected with the following criterion. Glycolysis/gluconeogenesis: an enzyme with irreversible reactions. PPP: genes involved in the main interconversion loop between ribose-5 phosphate and fructose-6 phosphate. Wood–Ljungdahl pathway: marker genes from Can et al.[61]. Methanogenesis: markers from Scheller et al.[41]. The glycosyl hydrolases were manually curated to assemble a set able to perform the substrate conversion.

**PECA analysis**. We ran PECA-R[46] to estimate translation and protein degradation rates using the absolute quantification tables for transcripts and proteins with default parameters. The rates are estimated between two consecutive time points, therefore the sample from 8 h was not included because it is missing the corresponding MT data. We filtered the results to identify the changing point using a score threshold of 0.9 and an FDR equal to 0.05.

**Reporting summary**. Further information on research design is available in the Nature Research Reporting Summary linked to this article.

## Data availability
All sequencing reads have been deposited in the sequence read archive (SRP134228), with specific numbers listed in Supplementary Table 6 in Kunath et al.[10]. All microbial genomes are publicly available on JGI under the analysis project numbers listed in Supplementary Table 6 in Kunath et al.[10]. The mass spectrometry proteomics data have been deposited to the ProteomeXchange Consortium via the PRIDE[62] partner repository with the data set identifier PXD016242. Source data in the form of quantification of transcript and protein levels are provided in Supplementary Data 3 and 4, respectively. Metabolomics quantifications are available at https://github.com/fdelogu/SEM1b-Multiomics/data.

## Code availability
The code used to perform the computational analysis is available at: https://github.com/fdelogu/SEM1b-Multiomics.

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

## Acknowledgements

We are grateful for support from The Research Council of Norway (FRIPRO program, PBP: 250479), as well as the European Research Commission Starting Grant Fellowship (awarded to P.B.P.; 336355 - MicroDE). The sequencing service was provided by the Norwegian Sequencing Centre (www.sequencing.uio.no), a national technology platform hosted by the University of Oslo and supported by the "Functional Genomics" and "Infrastructure" programs of the Research Council of Norway and the Southeastern Regional Health Authorities. P.N.E. is supported by an Australian Research Council Discovery Early Career Researcher Award (1700100428)

## Author contributions

F.D., T.R.H., and P.B.P. conceived the study, performed the primary analysis of the data, and wrote the paper (with input from all authors). B.J.K., and M.Ø.A. generated the data and contributed to the data analyses. P.N.E contributed to metabolic reconstructions for syntrophic acetate oxidizers and methanogens.

## Competing interests

The authors declare no competing interests.
