## [Peer Review File · Nature Communications]

REVIEWER COMMENTS

Reviewer #1 (Remarks to the Author):

The manuscript by Delogu et al. describes the measurement of absolute Protein/RNA ratios in microbial communities to measure functional community dynamics. For this the authors employ a combination of a quantitative metatranscriptomics and metaproteomics approach. Additionally the authors also provide measurements of key metabolites to integrate with the multi-omics dataset. The work represents what I would consider a major milestone in microbial ecology research and will likely have ramifications for many systems currently under study. The analyses were done with great care and the results are impressive and represented in very intuitive figures. The manuscript is well written and easy to follow. I do have some comments and concerns that if addressed will hopefully provide additional clarity in some parts.

Major comments:

1. In the abstract and throughout the manuscript you describe your measurements as “absolute RNA and protein levels”. However, absolute levels imply that copy numbers per cell or mass are given. What you actually present are “absolute Protein/RNA ratios”. I think this should be clarified in the abstract and throughout so that readers will not be disappointed when looking for absolute RNA numbers per cell or alike.
2. It is unclear in the methods how sample sizes for metatranscriptomics and metaproteomics were standardized. Since the Protein/RNA ratios are based on absolute molecule numbers per sample, it is critical to explain how you ensured that the samples were actually all the same “size” (cell mass, protein amount or alike).
3. I was unable to access the deposited proteomics raw data i.e. reviewer credentials were not provided. The data should be accessible to reviewers to check quality metrics.

Minor comments:

4. I think it would be good to integrate the “RNA-protein dynamics” or “Protein/RNA ratios” in the title somehow. The current title does not really indicate that this is the main parameter measured in this study.
5. Throughout the manuscript Bacteria, Archaea and Eukaryotes are classified as “kingdoms”, however, the more correct and current terminology is “domains”.
6. Lines 31/32: I am not sure I agree with this introductory sentence thinking of Winogradsky and van Leeuwenhoek at the foundation of microbiology (definitely not pure culture studies).
7. Line 48: I suggest replacing “prokaryotes” with “microorganisms” as MT and MP are also used to study eukaryotic microbes.
8. Lines 72/73: Unclear what “reconstruct at the molecular level” means.

9. Line 82: I could not find the details for the construction of ORF groups in the methods. The identity levels used should be mentioned here in the main text and the construction procedure elaborated on in the methods.
10. Line 89: replace “problematic” with “difficult to assemble”
11. Line 91: I do not understand the use of the word “speciation” here. Delete?
12. Line 103: Please re-state this to “...well-known technical issues with the gel-based sample preparation method that we used...” The current statement gives the incorrect impression that proteins with transmembrane domains are always hard to extract, however, FASP based and in solution methods do not have this issue as much see e.g. <https://pubs.acs.org/doi/10.1021/pr300709k>
13. Line 106: “slightly” and “moderately” are very unspecific. It would be helpful if you provided a sentence explaining how the test and the numbers are to be interpreted and how one can deduce the effect size can be seen based on them.
14. Line 116: This is the first instance where you mention your “per sample” measure of copy numbers. Here and in the methods you would need to explain what “sample” actually means and how this was standardized across the experiment.
15. Line 123: How was growth phase determined?
16. Lines 161-163: I find this statement to be one of the most important and central findings of your manuscript and am wondering if it would make sense to integrate it with your abstract.
17. Lines 184-186: Another results statement that is critical and that may be worthwhile mentioning in the abstract.
18. Line 241: Add to the end of the sentence “... with the gel-based method that we employed”
19. Figure 2: Great figure. The font on the y-axis is kind of small and hard to read. Is it grey instead of black?
20. Line 264: The abbreviation SAO was not introduced. I am wondering though if you really need to abbreviate this as it makes it hard for the reader to keep track of sooo many abbreviations. WLP could also be written out in the few instances it appears at.
21. Figure 3: Great figure.
22. Line 365: Replace “quantified the number of RNAs and proteins over time...” with “quantified absolute protein/RNA ratios...”
23. Lines 387 – 389: This is an interesting statement, the corresponding results&discussion section was not entirely clear on that, particularly the “degradation” part. It would be good to clarify the outcomes regarding degradation in the results and discussion.
24. Line 468: What type of beads were used for bead beating?

25. Line 471: Since protein quantification is critical for this study it would be good to provide additional details on Bradford assay replication.
26. Line 475: Describe the LC method used.
27. Line 480: I noticed that you use a resolution of 35K for MS2. I would recommend to decrease this value to the minimum in the future. At 35K the transient time in the Orbitrap is quite long and you will limit the number of MS2 events that you will be able to acquire per run. Since in MS2 pre-isolated ions are fragmented, high resolution is not needed as the spectra are low complexity and mass accuracy is barely affected by the resolution.
28. Line 491: The use of the “match between runs” feature seems risky in Metaproteomics and in particular for gel-based approaches as it is likely that incorrect mass peak identifications are transferred between runs. Have you tested this feature for metaproteomics?
29. Lines 527 and onward: Currently some of the parameters for the formula are not completely clear and more detail would be helpful for each parameter. What is for example “protein mass”? Is this the assay determined protein concentration? What are Totalproteinmass? Base peakintensity? Mass? Detectedproteinmass?
30. The “total protein approach” was developed using a gel-free, multi-enzyme protocol, while you used a gel-based, single enzyme approach. While I do not think that this will impact your main results and conclusions, I do think that it is critical for you to discuss potential caveats of your approach.

Signed: Manuel Kleiner

Reviewer #2 (Remarks to the Author):

In this paper, Delogu et al. quantified absolute RNA and protein levels per gene in a cellulose-degrading microbial consortium. With this data, the authors quantified the ratio of protein to RNA in members of the consortium and arrived at 10²-10⁴ protein molecules per RNA molecule for bacteria and roughly ten times that for archaea, replicating the ratio observed by Taniguchi et al. for the former and studies in eukaryotes for the latter. The authors calculated linearity (formally, the polynomial degree which best fits the relationship between protein and RNA) and used it to identify ecological relationships in the above consortium.

This manuscript seems rigorous, well-written and useful. Personally, I would have cited it in my latest paper had it been published. Below are a few comments, which, if addressed, I believe would increase the impact of this manuscript even further.

Major comments:

1. Grouping into ORFGs:

Any grouping and averaging reduces variation as it collapses a distribution into its summary statistic (whether you take a mean or median, doesn't matter). In this case, I wonder whether the grouping may have affected the variability in protein and RNA levels. I believe the authors only used singletons for downstream analysis (more on this to come), so protein/RNA ratio may not have been affected.

2. Using only singletons for downstream analyses:

Can the authors estimate the biases such a decision may cause? One thing I can think of is that some genes that are more common and conserved across organisms, and thus perhaps represent housekeeping functions, are more likely to be grouped, and therefore tossed before downstream analysis. This may skew the reported ratio of protein-to-RNA.

3. Abundance ranking analysis:

The authors report that membrane transport genes are poorly represented in MP (l. 102) and following that report some discrepancy between MT and MP and (a larger one) between MG and MP. I wonder if repeating this analysis with transport (and other membrane) genes removed would rescue the correlation and perhaps change the conclusion of this paragraph.

4. Timepoints:

In motivating the study, perhaps even in the introduction, it would help the general understanding of the manuscript if there was an illustration of the timeline and what each timepoint means. Especially if the authors can say which metabolites are present in the sample in each timepoint.

5. Reported values:

The medians reported in line 129-130 and those in Fig. 1a seem different. Also, Fig. 1a would better represent the data as a boxplot or violin plot.

6. PCC analysis:

Some of the conclusions that the authors get to from analyzing the correlation between protein and transcript may be premature. For example, intrinsic variability at the transcript level, say between replicates in each timepoint could explain the variability in protein/RNA ratio. Another question that arises is whether transcripts with higher expression are more or less variable in the protein/RNA ratio? The conclusion (not being able to predict) may not hold in some of these cases, and may not require a polynomial model to explain.

7. Gene group analysis:

I believe the manuscript would have a broader impact if the authors ask whether the protein/RNA ratio is higher/lower in specific gene groups? Is it more/less variable? Is there a difference between housekeeping and auxiliary genes? Not just in the context of cellulose metabolism, but in general. This could really shed light on stochasticity of gene expression and translation, and on places where there is a tradeoff between speed and stability (I think it was shown to an extent in Chapal et al. PLoS Biology 2019). I accept that this may be out of scope of this paper.

8. The use of “linearity” is misleading.

Linearity cannot be “good” (line 253); it just is. In the same way it cannot increase or decrease. Things are either linear, polynomial or sub-linear.

9. Phase considerations:

Does “translation control drive changes in cell status and resource utilization” as the section title suggests, or are these metrics driven by cell status? I would assume different values of “linearity” in different life stages of a microbial community, for example, if a community reaches stationary phase and some translation / transcription stops, the “linearity” would depend only on the half-life of protein or RNA molecules rather than affect the cell.

Minor comments:

Line 32 - “However, we are constantly told... “ - could use a reference.

Line 97 - KEGG should be all-caps as it is an acronym.

Line 116 - the s.d. seems excessive on an initial reading despite being not that bad. I'd elect to specify minimum and maximum levels instead (3.26×10^{11} - 6.06×10^{12} reads better and is more informative than specifying SD).

Line 128 - "949 being the most likely" is a misinterpretation. The mode is the most likely value, not the median.

Line 156 - "novel triphosphate structure" - novel how?

Line 164 - typo: microbiome's

Good luck and well done.

Reviewer #3 (Remarks to the Author):

Delogu et al. dissect a simplistic microbial consortium (SEM1B) using three orthogonal omics techniques – metagenomics, -transcriptomics, and -proteomics. Specifically, by profiling absolute levels of the individual biomolecules, they can uncover functional adaptations in individual consortium members over time, till an equilibrium is reached. This results in several interesting findings – some of which could not have been inferred from relative datasets, such as the fact that within the consortium bacterial cells contain approximately 1,000-fold more protein than RNA. Other findings, in contrast, could have also been deduced from relative measurements, e.g. bulk analyses of the expressed modules (Fig. 2) and – to some extent – even the finding that there is barely any correlation between mRNA and protein expression (albeit not in absolute, but in that case only in relative terms).

In general, this comprehensive study is relevant, timely, and technically well conducted. I have the following suggestions though, to further improve it.

- The authors should better carve out what specific benefits their absolute quantification has and which of their conclusions could have similarly been drawn from a relative quantification.

- According to their findings, there is little correlation between mRNA expression changes and the corresponding alterations on the protein level and it is thus “nearly impossible to predict the level of a given protein based on the level of the corresponding transcript” (see lines 184-186). Put provocatively, this raises the question as to why at all (meta-)transcriptomic experiments should be conducted. This is highly relevant for many researchers as RNA-seq is widely used and the authors should therefore provide here some guidelines as to when RNA-seq might still provide functional implications. (Or, in case they generally discourage from using RNA-seq for functional bacterial analyses, they should phrase it as such.)

- Some parts would benefit from a more detailed experimental description. For example, the authors should provide more experimental details of their metatranscriptomics analysis. For example, line 449 reads “After purification, residual DNA, free nucleotides and small RNAs were removed.” But it is not explained HOW this was achieved. Likewise, line 450: “Samples were treated to enrich for mRNAs (...)” Here again, how this was done is not mentioned. Further, I’d appreciate if the authors compiled a supplementary table with the mapping statistics of the metatranscriptomics data (number of reads/sample; percentage of mapped vs. unmapped reads/sample; distribution of the mapped reads to their respective source genomes; etc.). This would also help the reader to obtain an idea as to how the relative composition of the consortium changes over time (or if it remains unchanged).

The overall experimental design is still unclear to me: In lines 430-432 it is stated that “The time series analyses consisted of metabolomics, metaproteomics and metatranscriptomics over nine time points (...) in triplicate”. However, reading on it sounds like not all time points of this timecourse were analyzed by all three omics approaches. Could the authors please clarify? In general, a supplementary figure showing a scheme of the samples taken and indicating with which omic method they were analyzed would help the reader to better appreciate their study.

Also in the methods section, the term “as previously described” should be avoided; rather, the experiment should be fully described in the current manuscript (I believe this is anyways an author guideline given by the journal).

Additional, minor comments include:

- Line 89: Change “algorithms has” to “algorithms have”.

- Line 158: “RNA is regulated by post-translational modifications of the RNA molecule”  Do the authors mean post-TRANSCRIPTIONAL modifications?

- Line 201: “start at values between 0.6 and 0.8 at 13 hours”  Please rephrase as there are clearly values outside this range in Fig. 1d (also for non-TEPI2 MAGs).

- Lines 239-240: “Notably from Fig. 2, it is clear that the proteins from the transporters are almost never found in the samples, even if the respective RNAs are abundant.”  As far as I understand, the discrepancy between RNA and protein level detection cannot be deduced from Fig. 2.
- Fig. 3 b-d: The units for the values plotted on the y-axes are missing (also not mentioned in the corresponding figure legend).
- Line 336: “in bacteria is believed to occur predominantly via transcription control (...)”  The authors may want to rephrase this. This concept has been overhauled in the past decade, realizing the widespread post-transcriptional control mechanisms – brought about by regulatory, noncoding RNAs – across the bacterial phylogenetic tree.
- Line 511 (and elsewhere): Please define what “Nt” refers to in this context.
- Suppl. Fig. S1: The x-axis for the metabolomics bar chart lacks any values.

**COMMENTS FROM REVIEWER(S):**

**Referee #1 (Comments to the Author):**

The manuscript by Delogu et al. describes the measurement of absolute Protein/RNA ratios in
microbial communities to measure functional community dynamics. For this the authors employ
a combination of a quantitative metatranscriptomics and metaproteomics approach.
Additionally the authors also provide measurements of key metabolites to integrate with the
multi-omics dataset. The work represents what I would consider a major milestone in microbial
ecology research and will likely have ramifications for many systems currently under study. The
analyses were done with great care and the results are impressive and represented in very
intuitive figures. The manuscript is well written and easy to follow. I do have some comments
and concerns that if addressed will hopefully provide additional clarity in some parts.

**RESPONSE:** We thank the reviewer for their comments and suggestions, which we have responded
to in full below.

**Major remark:**

**1. In the abstract and throughout the manuscript you describe your measurements as “absolute**
**RNA and protein levels”. However, absolute levels imply that copy numbers per cell or mass**
**are given. What you actually present are “absolute Protein/RNA ratios”. I think this should**
**be clarified in the abstract and throughout so that readers will not be disappointed when**
**looking for absolute RNA numbers per cell or alike.**

**RESPONSE:** We agree with the reviewer that this study focusses on ‘**absolute protein/RNA ratios**’
and have therefore adapted the term throughout the manuscript, including the title as suggested below
(**Reviewer#1_Q4**). We measured the “absolute RNA and protein levels” at the total sample level (i.e.
total consortium), similarly to the normalized quantities that were normalized at the consortium level
but not at the population level (e.g. TPMR, PTM, LFQ). We agree that computing the copy number of
molecules per cell would have been interesting and indeed we thought to adapt the strategy proposed
in the “total protein approach” paper to a microbial community setting. However, the presence of
insoluble and dense particulate matter in our media (i.e. lignocellulose) proved unamendable to
estimate cell numbers via direct counting, whereas our mixed inter-dependent consortia prohibited
statistical counting methods (e.g. most probable number).

**2. It is unclear in the methods how sample sizes for metatranscriptomics and metaproteomics**
**were standardized. Since the Protein/RNA ratios are based on absolute molecule numbers**
**per sample, it is critical to explain how you ensured that the samples were actually all the**
**same “size” (cell mass, protein amount or alike).**

**RESPONSE:** We thank the reviewer for pointing out this important issue. We changed the subheading
in **Line 461** from “***Background***” to “***Background and multi-omics sampling***”. In addition, in the same
section we added the following text:

**Line 467:** “*For every time point (60 ml), a 6 ml and 30 ml aliquot were collected and used for MT and*
*MP analysis, respectively. The RNA internal standard was added to the 6ml aliquot and the resulting*
*transcript levels were therefore multiplied by 10 to reconstruct the original 60 ml sample size (3x*
*replicates). In case of the MP analysis, the proteins were extracted from the 30 ml aliquot and the*
*protein concentration calculated using the Bradford method from which we computed the original mass*
*of protein in the 60 ml sample (3x replicates). Therefore, the number of transcripts and proteins used*
*in the paper refer to the whole consortium contained within each culture flask.”*

**3. I was unable to access the deposited proteomics raw data i.e. reviewer credentials were not**
**provided. The data should be accessible to reviewers to check quality metrics.**

**RESPONSE:**
We apologize for not including the login information to the PRIDE repository. Please use the
following:

<https://www.ebi.ac.uk/pride/archive/login>

Username: reviewer35204@ebi.ac.uk

Password: WWZ9gRiC1

**Minor remarks:**

**4. I think it would be good to integrate the “RNA-protein dynamics” or “Protein/RNA ratios”**
**in the title somehow. The current title does not really indicate that this is the main parameter**
**measured in this study.**

**RESPONSE:** We have changed the title to “*Integration of absolute multi-omics reveals dynamic*
*protein-to-RNA ratios and metabolic interplay within mixed-domain microbiomes*”

5. Throughout the manuscript Bacteria, Archaea and Eukaryotes are classified as “kingdoms”, however, the more correct and current terminology is “domains”.

RESPONSE: The error has been corrected throughout the paper, including the title (see **Reviewer#1_Q4** above).

6. Lines 31/32: I am not sure I agree with this introductory sentence thinking of Winogradsky and van Leeuwenhoek at the foundation of microbiology (definitely not pure culture studies).

RESPONSE: We agree with the reviewer’s statement that within the context of microbiology’s founders Winogradsky and van Leeuwenhoek our original sentence is not optimal, and have replaced the word “foundations” with “*fundamentals*”

7. Line 48: I suggest replacing “prokaryotes” with “microorganisms” as MT and MP are also used to study eukaryotic microbes.

RESPONSE: We corrected the wording as suggested.

8. Lines 72/73: Unclear what “reconstruct at the molecular level” means.

RESPONSE: We mean to characterize (and quantify) the molecular components of the community. The sentence has been modified to:

Line 75: *“In order to explore the RNA/protein dynamics in a microbiome setting, we first needed to characterize our test community over time at the molecular level.”.*

9. Line 82: I could not find the details for the construction of ORF groups in the methods. The identity levels used should be mentioned here in the main text and the construction procedure elaborated on in the methods.

RESPONSE: We did not compute the ORF groups explicitly but used tools for quantification of MT (mmseq + mmcollapse) and MP (MaxQuant) that perform the grouping. To improve the clarity of our paper, we have added a sentence in the main text to redirect the reader to the methods and added the following sentences in the methods:

**Line 85:** “*Since ORFs with very high sequence similarity may produce RNAs and proteins that are*
*indistinguishable in MT and MP data, all the ORFs were gathered into ORF-groups (ORFGs) during*
*the MT and MP data processing (see methods), where a singleton ORFG is defined as a group with a*
*single ORF, and thus a single gene.*”

**Line 532:** “*The collapse first gathers ORFs into groups if they have 100% sequence identity, and in a*
*second round the ORFs (already termed ORFGs) are collapsed if they acquire unique hits as a group.*”

**Line 582:** “*When proteins cannot be unambiguously identified with unique peptides, MaxQuant will*
*group them and quantify them together as one ORFG.*”

**10. Line 89: replace “problematic” with “difficult to assemble”**

**RESPONSE:** We changed the wording as suggested.

**11. Line 91: I do not understand the use of the word “speciation” here. Delete?**

**RESPONSE:** We refer to the process of divergence of two species from an original one. We changed
the wording to “*species divergence*”.

**12. Line 103: Please re-state this to “...well-known technical issues with the gel-based sample**
**preparation method that we used...” The current statement gives the incorrect impression**
**that proteins with transmembrane domains are always hard to extract, however, FASP**
**based and in solution methods do not have this issue as much see e.g.**
**<https://pubs.acs.org/doi/10.1021/pr300709k>**

**RESPONSE:** We agree with the reviewer and have changed this section as suggested:

**Line 118:** “*The Membrane transport category is poorly represented in the MP (2% of the terms), which*
*is likely explained by well-known technical issues with the gel-based sample preparation method that*
*we used, which limits the extraction of transmembrane proteins¹⁸.*”

**13. Line 106: “slightly” and “moderately” are very unspecific. It would be helpful if you**
**provided a sentence explaining how the test and the numbers are to be interpreted and how**
**one can deduce the effect size can be see based on them.**

**RESPONSE:** In response to this comment, we have added two sentences on the Kendall τ and its
interpretation, alongside a rephrase to present better the results:

**Line 121:** *“The abundance ranking of the KO categories was assessed using the Kendall τ , which takes*
*values from -1 (opposite direction of the ranking) to +1 (total agreement in ranking). Its score is*
*interpreted as a correlation measure; however, it is more conservative. The ranking is largely*
*preserved from MG to MT (Kendall τ : 0.77, $p < 10^{-8}$) and from MT to MP (τ 0.74, $p < 10^{-6}$) whilst less so*
*from MG to MP (τ 0.68, $p < 10^{-5}$). The results show that the functional potential observed in the genomes*
*is more preserved in the diversity of produced transcripts than in the produced proteins and thus hints*
*to post-transcriptional regulation playing an important role in addition to transcriptional regulation*
*in prokaryotes.”*

**14. Line 116: This is the first instance where you mention your “per sample” measure of copy**
**numbers. Here and in the methods you would need to explain what “sample” actually means**
**and how this was standardized across the experiment.**

**RESPONSE:** As requested above in **Reviewer#1_Q2**, we added the explanation of the sample
normalization. In addition, we have modified the following sentence:

**Line 133** *“[...] metabolic states and/or taxonomic phylogeny, we quantified and resolved the numbers*
*of transcript and protein molecules per sample (i.e. the total SEM1b consortia within each 60ml flask,*
*see Material and Methods), which averaged 3.8×10^{12} (sd 3.0×10^{12}) and 2.2×10^{15} (sd 9.5×10^{14}),*
*respectively (Supplementary Datasets 3-4).”*

**15. Line 123: How was growth phase determined?**

**RESPONSE:** Given the aforementioned difficulties associated with cell counting (e.g. high levels of
insoluble material: see **Reviewer#1_Q1**), we used the amount of protein production as a proxy for the
overall community growth curve as in our previous work (**Kunath 2019**, Fig. 5A). This data has now
been integrated into a newly created **Figure 1**, which also addresses several other reviewer comments

concerning the experimental design used in this study (Reviewer#2_Q4) and the sampling scheme
 (Reviewer#3_Q3).

 **16. Lines 161-163: I find this statement to be one of the most important and central findings of**
 **your manuscript and am wondering if it would make sense to integrate it with your abstract.**
 **RESPONSE:** Sentence: *“In a microbiome-setting, the greater turnover of RNA molecules and lower*
 *protein-RNA ratio in bacteria could potentially facilitate their faster adaption to changes in metabolic*
 *state and substrate availabilities in their environment, at higher rates than their archaeal*
 *counterparts.”*
 We agree on the importance of the finding in light of microbial ecology; however, we feel that this
 statement is perhaps too speculative to include in the abstract given our experimental design did not
 include perturbations that could test for such a rapid adaptation. In addition, the abstract is subjected to
 strict space constraints (just 150 words), we decided to emphasize the numerical and modelling aspects
 of the work and postponing the implications to the main text.

**17. Lines 184-186: Another results statement that is critical and that may be worthwhile**
**mentioning in the abstract.**

**RESPONSE:** Sentence: *“This suggested that no direct correlations between RNA and proteins levels*
*exist at any stage at a community level and that it is nearly impossible to predict the level of the given*
*protein based on the level of the corresponding transcript.”*

These aforementioned observations have been previously reported in *E.coli* by **Taniguchi 2010** et al,
and our results confirm this at a larger community level. Therefore, we feel that this slightly diminishes
the novelty of this statement, and hence we chose to use the very limited word count in the abstract to
focus on the discrepancy between bacteria and archaea and the relationship between the protein/RNA
ratio and population function.

**18. Line 241: Add to the end of the sentence “... with the gel-based method that we employed”**

**RESPONSE:** We have added the suggested text.

**19. Figure 2: Great figure. The font on the y-axis is kind of small and hard to read. Is it grey**
**instead of black?**

**RESPONSE:** Yes, it was grey. We changed the color of the font on the y-axis to black and increased
the size to make it easier to read.

**20. Line 264: The abbreviation SAO was not introduced. I am wondering though if you really**
**need to abbreviate this as it makes it hard for the reader to keep track of sooo many**
**abbreviations. WLP could also be written out in the few instances it appears at.**

**RESPONSE:** We agree about improving readability and have therefore expanded all the instances of
“SAO” to “syntrophic acetate oxidizing/ation” and “WLP” to “Wood-Ljungdahl Pathway”.

**21. Figure 3: Great figure.**

**RESPONSE:** Thanks!

**22. Line 365: Replace “quantified the number of RNAs and proteins over time...” with**
**“quantified absolute protein/RNA ratios...”**

**RESPONSE:** In line with the answer to the above comment (**Reviewer#1_Q1**) we replaced
“*quantified the number of RNAs and proteins over time...*” with “*quantified the number of RNAs and*
*proteins per sample over time in absolute terms and as ratios*”.

**23. Lines 387 – 389: This is an interesting statement, the corresponding results&discussion**
**section was not entirely clear on that, particularly the “degradation” part. It would be good**
**to clarify the outcomes regarding degradation in the results and discussion.**

**RESPONSE:** We agree our use of PECA-R to estimate the predict the change of protein translation
and/or degradation rate is not clear. In response to this comment we have included some additional text
in the results and discussion section to help improve the clarity of our use of the term “degradation”:

**Line 363:** “*We used our absolute quantifications of SEM1b transcripts and proteins as well as PECA-*
*R⁴⁴ to predict the “change-point”, which takes into account estimates of protein translation and*
*degradation rates (Supplementary Dataset 5).*”

**24. Line 468: What type of beads were used for bead beating?**

**RESPONSE:** We used glass beads (size, $\leq 106 \mu\text{m}$). In **Line 541** we have changed the sentence “*Cells*
*were disrupted in 3×60 seconds cycles using a FastPrep24 (MP Biomedicals, USA) [.]*” adding: “[...]”
*with glass beads (size, $\leq 106 \mu\text{m}$).*” at the end.

**25. Line 471: Since protein quantification is critical for this study it would be good to provide**
**additional details on Bradford assay replication.**

**RESPONSE:** We have additional details for the Bradford assay as requested:

**Line 544:** “*Extracted proteins were quantified using the Bradford’s method (in triplicate), which*
*quantified the samples by combining 2 to 10 μl of protein extract with 20mM Tris HCL (pH 7.5) to*
*reach 800 μl , with 200 μl of BioRad Essay solution subsequently added. Samples were vortexed,*
*centrifuged briefly and let to rest for 5 minutes before measuring with dedicated cuvettes. Blanks*
*composed of 800 μl of buffer and 200 μl of BioRad Essay solution were used before each set of*
*measurements.*”

**26. Line 475: Describe the LC method used.**

**RESPONSE:** We have included the following more in-depth description as requested:

**Line 559:** *“Peptides were analyzed using a nanoLC-MS/MS system consisting of a Dionex Ultimate*
*3000 UHPLC (Thermo Scientific, Germany) connected to a Q-Exactive hybrid quadrupole-orbitrap*
*mass spectrometer (Thermo Scientific, Germany) equipped with a nanoelectrospray ion source. The*
*samples were loaded onto a trap column (Acclaim PepMap100, C18, 5 μm , 100 \AA , 300 μm i.d. x 5 mm,*
*Thermo Scientific) and back flushed onto a 50-cm analytical column (Acclaim PepMap RSLC C18, 2*
*μm , 100 \AA , 75 μm ID, Thermo Scientific). At the start, the columns were in 96% solution A [0.1% (v/v)*
*formic acid], 4% solution B [80% (v/v) acetonitril, 0.1% (v/v) formic acid]. The peptides were eluted*
*using a 90 minutes gradient developing from 4% to 13% (v/v) solution B in 2 minutes, 13% to 45%*
*(v/v) B in 70 minutes and finally to 55% B in 5 minutes before the wash phase at 90% B, all at a flow*
*rate of 300 nL/min.”*

**27. Line 480: I noticed that you use a resolution of 35K for MS2. I would recommend to decrease**
**this value to the minimum in the future. At 35K the transient time in the Orbitrap is quite**
**long and you will limit the number of MS2 events that you will be able to acquire per run.**
**Since in MS2 pre-isolated ions are fragmented, high resolution is not needed as the spectra**
**are low complexity and mass accuracy is barely affected by the resolution.**

**RESPONSE:**

We thank the Reviewer for this observant comment. We will indeed check our methods and reduce
the MS/MS resolution to 17.500 for future samples.

**28. Line 491: The use of the “match between runs” feature seems risky in Metaproteomics and**
**in particular for gel-based approaches as it is likely that incorrect mass peak identifications**
**are transferred between runs. Have you tested this feature for metaproteomics?**

**RESPONSE:** “Match between runs” (MBR) is currently the *modus operandi* for label-free MS1
quantification in proteomics and incorporated in many software such as MaxQuant, moFF and
FlashLFQ. The feature enables identification of peptides without an MS/MS spectrum and instead
using the accurate m/z and retention time of the peptide, if this aligns with another peptide with the
same mass and retention time and with a confident MS/MS identification in another run. This has been

thoroughly described and evaluated in several publications (e.g. **Lim 2019** and **Cox 2014**) and the
conclusion is that the number of false IDs due to MBR is low (~1-3%).

However, the evaluation in previous publications have used single-species proteomes and we share
the reviewer's concern that if samples are very rich, which is typical for metaproteomics, this could
yield a higher level of incorrect identifications. We have unfortunately not tested this specifically for
metaproteomics, but we want to stress that we performed gel-separation of proteins into 16 fractions
269 per sample prior to MS to simplify the peptide mixtures. Further, the MBR only applies to the gel-
270 fractions and their adjacent fractions, e.g. fraction five in lane A will only be compared to fraction
four, five and six in lane B, etc., so not everything against everything. This means that as long as
extensive gel-separation is performed (here 16 fractions per lane), and the chromatography is stable
and the mass spectrometer is accurate, which all is the case for these samples, the potential of
erroneous identification should remain low in our opinion.

**29. Lines 527 and onward: Currently some of the parameters for the formula are not completely**
**clear and more detail would be helpful for each parameter. What is for example "protein**
**mass"? Is this the assay determined protein concentration? What are Totalproteinmass?**
**Base peakintensity? Mass? Detectedproteinmass?**

**RESPONSE:** We improved the name of the parameters by adding spaces where required as well as
expanding the section and defining all the mentioned parameters:

**Line 632:** *"Knowing the Total protein mass [g] per sample, computed from the protein concentration*
*estimated with the Bradford assay and the starting volume of the sample (60 ml), we estimated the*
*detected protein mass [g] (i.e. how much of protein mass is explained by the peaks recognized during*
*MP analysis) using the raw MP files with the following formula:*

$$287 \quad \text{Detectedproteinmass} = \frac{\text{Totalproteinmass} \times \sum_{i=1}^{\text{Pepid.}} \text{Base peakintensity}_i \times \text{Mass}_i}{\sum_{j=1}^{\text{Pep}_{\text{tot}}} \text{Base peakintensity}_j \times \text{Mass}_j}$$

*Where the "Base peak intensity_i" and "Mass_i" corresponds to the homologous values for to the ⁱth*
*identified peak (i.e. a peak with an amino acid sequence associated, thus it can be called a peptide) in*
*the raw MP files."*

**30. The “total protein approach” was developed using a gel-free, multi-enzyme protocol, while**
**you used a gel-based, single enzyme approach. While I do not think that this will impact your**
**main results and conclusions, I do think that it is critical for you to discuss potential caveats**
**of your approach.**

**RESPONSE:** The Reviewer is indeed correct that there are reports suggesting that there may be a
protease bias in absolute protein quantification (e.g. **Peng 2012**) pointing to irregularity in peptide
formation and that using >1 protease provides a more robust assessment of the protein abundance. This
is not an issue with relative quantifications as the same protease is used for all conditions tested but
may affect the protein abundance when absolute values are used. This issue was tested by Wiśniewski
when developing the ‘total protein approach’ (**Wiśniewski 2014**) where they compared a pure tryptic
digest with a combined LysC+Trypsin digestion. In contrast to the first report, Wiśniewski showed a
high correlation in protein quantification between these two digestion strategies ($R^2=0.88-0.92$),
suggesting that a pure tryptic approach may be sufficient. To make the reader aware of the potential
caveats of our approach, we have included the following lines:

**Line 552:** *“It is important to note that previous reports have shown that absolute protein quantification*
*may be biased by the protease selected for digestion (**Peng 2012**). While ideally, more than one*
*protease could have been used, we have used trypsin in our analysis (which is commonplace in most*
*proteomics experiments) and obtained high correlation of absolute protein quantification between*
*replicates (average $R^2=0.85$, with the outlier t7C removed). Moreover, the protein-to-RNA ratios*
*observed for the different bacteria and archaea correlate well with previous literature (*E. coli*, yeast,*
*human), indicating that our absolute quantifications are on par.”*

**Referee #2 (Comments to the Author):**

**In this paper, Delogu et al. quantified absolute RNA and protein levels per gene in a cellulose-**
**degrading microbial consortium. With this data, the authors quantified the ratio of protein to**
**RNA in members of the consortium and arrived at 10^2-10^4 protein molecules per RNA molecule**
**for bacteria and roughly ten times that for archaea, replicating the ratio observed by Taniguchi**
**et al. for the former and studies in eukaryotes for the latter. The authors calculated linearity**

(formally, the polynomial degree which best fits the relationship between protein and RNA) and
used it to identify ecological relationships in the above consortium.

This manuscript seems rigorous, well-written and useful. Personally, I would have cited it in my
latest paper had it been published. Below are a few comments, which, if addressed, I believe
would increase the impact of this manuscript even further.

**RESPONSE:** We thank the reviewer for their positive and encouraging comment!

**Major remarks:**

**1. Grouping into ORFGs:**

Any grouping and averaging reduces variation as it collapses a distribution into its
summary statistic (whether you take a mean or median, doesn't matter). In this case, I
wonder whether the grouping may have affected the variability in protein and RNA levels.
I believe the authors only used singletons for downstream analysis (more on this to come),
so protein/RNA ratio may not have been affected.

**RESPONSE:**

In general, when it is impossible to distinguish two or more genes' products because they share
common hits (MT reads or peptides) without a single unique hit, we are faced with three main
choices: i) remove the shared genes' products from the dataset; ii) distribute the hits; iii) group the
genes' products and assign the entire pool of hits to the group. The first option results in the loss of
information, the second one lacks a fair criterion to distribute the reads (because no unique hits are
present) and the last one loses resolution. Of the three we preferred option iii), in order to use all the
data available and not over/under-represent any gene product using an arbitrary rule to split the hits
(e.g. using the average).

However, we agree and share the same opinion as the reviewer. Indeed we used only the singleton set
in all the analyses that required direct comparison of ORFs, such as transcript-protein correlation and
protein-to-RNA ratios. We added this second example in the Methods section, which now reads as
follows:

Line 647: “When the analysis required the direct comparison of ORFs (e.g. transcript-protein
correlation and protein-to-RNA ratios) only the singleton subset of the ORFGs was considered.”

2. Using only singletons for downstream analyses:

Can the authors estimate the biases such a decision may cause? One thing I can think of is
that some genes that are more common and conserved across organisms, and thus perhaps
represent housekeeping functions, are more likely to be grouped, and therefore tossed
before downstream analysis. This may skew the reported ratio of protein-to-RNA.

**RESPONSE:** The reviewer raises an interesting comment regarding the conservation of gene
sequences and functions across different microbial populations. If we consider the relatedness of the
populations analyzed in this study, we see a broad representation of phylogenetic diversity, with a
total of 2 domains, 3 phyla and 5 different orders represented in our SEM1b consortium (TEPI1 and
TEPI2: *Thermoanaerobacterales*, RCLO1, CLOS1 and TISS1: *Clostridiales*, COPR1:
*Coprothermobacterota*, METH1: Euryarchaeota) (see Fig. below from <https://pubmed.ncbi.nlm.nih.gov/30315317/>).

If ORFs that are conserved across all SEM1b populations are being removed from our analysis at a
level that would inflict bias in our protein/RNA ratio estimates, we would have expected to observe
variations between taxa that consisted of multiple closely related populations (i.e. COPR1) and those
that are distinct (i.e. TISS1). Indeed, we observed no substantial variations across the protein/RNA
ratio in SEM1b populations, and those from a pure culture study of *E. coli* that is subjected to no such
issue of bias (see Fig. 2a), which leads us to believe the effect of conserved ORFs is not overly
influencing our results. We strongly agree that the variation in protein/RNA ratio within given
function categories (i.e. housekeeping, auxiliary, etc...) warrants further investigations, however, feel
it is outside the scope given the size of this task in context to what our manuscript has already
contributed.

3. Abundance ranking analysis:

**The authors report that membrane transport genes are poorly represented in MP (l. 102)**
**and following that report some discrepancy between MT and MP and (a larger one)**
**between MG and MP. I wonder if repeating this analysis with transport (and other**
**membrane) genes removed would rescue the correlation and perhaps change the**
**conclusion of this paragraph.**

**RESPONSE:** We thank the reviewer for the interesting question. In response to this comment we
repeated the analysis removing the “Membrane transport” category, however the resulting values for
Kendall τ are similar to the original ones, meaning we can infer that the divergence between the
omics layers generalizes beyond the (large) discrepancy observed in this single functional category.

	MG-MT	MT-MP	MG-MP
“Membrane transport” included	0.77	0.74	0.68
“Membrane transport” excluded	0.76	0.74	0.68

4. Timepoints:

**In motivating the study, perhaps even in the introduction, it would help the general**
**understanding of the manuscript if there was an illustration of the timeline and what each**

timepoint means. Especially if the authors can say which metabolites are present in the
sample in each timepoint.

**RESPONSE:** We agree, and in response we created a new **Figure 1** (illustrated above in response to
**Reviewer#1_Q15**) to illustrate the life arch of the microbial community and the main events related
to metabolites over time, whilst referring to the individual metabolite plots in **Figure 4** (old Figure
3). The new Figure 1 also integrates the request to depict the growth curve (**Reviewer#1_Q15**) and
illustrate the sampling scheme (**Reviewer#3_Q3**).

5. Reported values:

The medians reported in line 129-130 and those in Fig. 1a seem different. Also, Fig. 1a
would better represent the data as a boxplot or violin plot.

**RESPONSE:** Thank you for this comment! By mistake we plotted the values relative to t2 instead of
t3 (which the numbers in the text referred to). We redid the plot as a boxplot (see below this
comment). This version is now panel a of **Fig. 2** (old **Fig. 1**).

**6. PCC analysis:**

**Some of the conclusions that the authors get to from analyzing the correlation between**
**protein and transcript may be premature. For example, intrinsic variability at the**
**transcript level, say between replicates in each timepoint could explain the variability in**
**protein/RNA ratio. Another question that arises is whether transcripts with higher**
**expression are more or less variable in the protein/RNA ratio? The conclusion (not being**
**able to predict) may not hold in some of these cases, and may not require a polynomial**
**model to explain.**

**RESPONSE:** We share the interests of the reviewer on the topic of noise (i.e. intrinsic variability) and
the dependence of the RNA/protein dynamics from transcript levels. In **Taniguchi 2010** these aspects
were addressed at the single-cell level and shown to be important factors in determining the lack of
correlation between protein and RNA level in the cell. When **Taniguchi 2010** compared directly
transcript and protein levels, it was done with the averaged values from all the sampled cells in order
to remove those effects. Therefore, we believe that our data, which concern the number of molecules
*per sample*, should not be subject to this phenomenon. Moreover, the average R^2 of the MT replicates,
excluding the outlier *t7C*, is 0.85; which indicates that the absolute MT values are highly correlated
within each time point. In order to increase the clarity of the results we included the following sentence
in the text:

**Line 201:** *“A high average R^2 value (0.85 for both MT and MP) was also determined between replicates*
*indicating the stability of our results and the lack of influence from random noise.”*

**7. Gene group analysis:**

**I believe the manuscript would have a broader impact if the authors ask whether the**
**protein/RNA ratio is higher/lower in specific gene groups? Is it more/less variable? Is there**
**a difference between housekeeping and auxiliary genes? Not just in the context of cellulose**
**metabolism, but in general. This could really shed light on stochasticity of gene expression**
**and translation, and on places where there is a tradeoff between speed and stability (I think**
**it was shown to an extent in Chapal et al. PLoS Biology 2019). I accept that this may be out**
**of scope of this paper.**

**RESPONSE:** This is an interesting question in which we are already moving, yet we feel that the size
of the task that the reviewer proposes in context to the novelty this current paper has already contributed
as well as new hypotheses proposed, makes this out of the scope of this paper (as the reviewer nicely
points out!).

**8. The use of “linearity” is misleading.**

**Linearity cannot be “good” (line 253); it just is. In the same way it cannot increase or**
**decrease. Things are either linear, polynomial or sub-linear.**

**RESPONSE:** We agree and changed the text in order to refer to the k value and its changes:

**Line 664:** *“The slopes of the models were then used to fit a third-grade polynomial function to obtain*
*the k value change profile in Fig. 2d.”*

**Line 220:** *“The evolution of the MAGs’ k values over time is then divided in three groups: one where*
*the k values decrease rapidly (TISS1 and COPR1); one where they slowly decline (RCLO1, CLOS1*
*and METH1) and one where they stay constant if not increase (TEPI1 and TEPI2) (Fig. 2d). Notably*
*CLOS1, METH1 and TEPI1 are converging towards the same k values [...].”*

**Line 292:** *“While TISS1 seems mostly to phase out of the community and its k value associated to its*
*protein to transcript relationship (Fig. 2d), [...].”*

**Line 273:** *“[...] as it demonstrated high k values that increased over time [...].”*

**Line 410:** *“In addition, we assessed the k values (proxy for linearity) associated to transcriptome and*
*proteome for each population over time (Eq.1), finding that three major populations of the community,*
*a fermenter (CLOS1), a syntrophic acetate oxidizing bacterium (TEPI1) and a methanogen (METH1),*
*were converging on the same values in parallel with the primary cellulose degrader (RCLO1) (Fig.*
*2d).”*

**9. Phase considerations:**

**Does “translation control drive changes in cell status and resource utilization” as the**

section title suggests, or are these metrics driven by cell status? I would assume different
values of “linearity” in different life stages of a microbial community, for example, if a
community reaches stationary phase and some translation / transcription stops, the
“linearity” would depend only on the half-life of protein or RNA molecules rather than
affect the cell.

**RESPONSE:** We decided to use the k values as a proxy for how close a population was to its steady
state, exploiting the fact steady state is reached when a change in protein level is mainly explained by
a change in transcript level (Liu 2016). When considering the change in translational control instead,
we were targeting the individual genes and their functions. Indeed, when we refer to “*drive changes*
*in cell status and resource utilization*” our intention is to discuss changes at a more metabolic and
lifestyle level. A change as such would be the switch between two substrates or the trigger to produce
spores. Probably both cases will be reflected in the k value associated to that population and we can
infer from it if it is approaching or steering away from the steady state. In the example from the
reviewer we would probably see certain cell functions being activated and other being shut down
which will be testified by a lower k value. To improve clarity and integrate the discussion risen by
the reviewer’s comment we have integrated the following sentences in the text:

**Line 354:** “*A change in protein regulation can be causally linked to a change in the population status*
*(steady state or transition). Within the cell, proteins are predominately the performers of cellular*
*functions thus the change in cell status can be achieved by actively altering the protein level.*”

**Minor remarks:**

**10. Line 32 - “However, we are constantly told... “ - could use a reference.**

**RESPONSE:** We have included the reference Palkova 2004

**11. Line 97 - KEGG should be all-caps as it is an acronym.**

**RESPONSE:** Corrected as requested.

**12. Line 116 - the s.d. seems excessive on an initial reading despite being not that bad. I'd elect**
**to specify minimum and maximum levels instead (3.26×10^{11} - 6.06×10^{12} reads better and is**
**more informative than specifying SD).**

**RESPONSE:** We agree, and we changed the sentence to:

**Line 135:** “[...] which averaged 3.8×10^{12} (range 3.45×10^{11} - 1.10×10^{13}) and 2.2×10^{15} (range
2.88×10^{14} - 3.46×10^{15}), respectively (Supplementary Datasets 3-4).”

**13. Line 128 - “949 being the most likely” is a misinterpretation. The mode is the most likely
value, not the median.**

**RESPONSE:** We agree and removed “being the most likely”.

**14. Line 156 - “novel triphosphate structure” - novel how?**

**RESPONSE:** We feel that this statement of ‘novelty’ is incorrect and has been amended to suggest
that 5’-triphosphate ends of the mRNA has a cap structure similar to eukaryotes which provides greater
resistance to mRNA degradation which is more aligned with what reference 26 refers to. Therefore,
the following sentence has been amended:

**Line 172:** “Correspondingly, the RNA of Eukarya and Archaea have been shown to exhibit longer half-
lives than Bacteria^{24, 25}, with some Archaea found to possess a cap complex similar to those in
eukaryotes at the 5’-triphosphate end of the RNA molecule that correlates with increased mRNA
stability²⁶”.

**15. Line 164 - typo: microbiome’s**

**RESPONSE:** Corrected.

**Referee #3 (Comments to the Author):**

**Delogu et al. dissect a simplistic microbial consortium (SEM1B) using three orthogonal omics**
**techniques – metagenomics, -transcriptomics, and -proteomics. Specifically, by profiling absolute**
**levels of the individual biomolecules, they can uncover functional adaptations in individual**
**consortium members over time, till an equilibrium is reached. This results in several interesting**
**findings – some of which could not have been inferred from relative datasets, such as the fact that**

within the consortium bacterial cells contain approximately 1,000-fold more protein than RNA.
Other findings, in contrast, could have also been deduced from relative measurements, e.g. bulk
analyses of the expressed modules (Fig. 2) and – to some extent – even the finding that there is
barely any correlation between mRNA and protein expression (albeit not in absolute, but in that
case only in relative terms).

In general, this comprehensive study is relevant, timely, and technically well conducted. I have
the following suggestions though, to further improve it.

**Major remark:**

**1. The authors should better carve out what specific benefits their absolute quantification has**
**and which of their conclusions could have similarly been drawn from a relative**
**quantification.**

**RESPONSE:** We agree that relative quantification of omic data is much more commonly used and
reported in microbiology studies and would have largely revealed the same changes in expression
patterns that were highlighted in **Figure 4**. However, the use of absolute quantification allows to have
a measurement that is sample- and experiment-independent and can be directly compared with other
samples and studies. In addition, it bypasses the compositionality problem (the sum of a percentage is
a fixed quantity) and in case of our specific method, it can unlock detailed quantitative knowledge of
biological systems, which was before out of range (cost- and labor-wise) for most of the laboratories.
In response to this comment, we have included some additional discussion to convey this benefit:

**Line 395:** *“In addition, relative quantification of omic data is much more commonly used and reported*
*in microbiology studies and would have largely revealed the same changes in expression patterns that*
*were highlighted in **Figure 4**. However, our absolute approach enabled us to assess and report, for*
*the first time, the protein-to-RNA ratio of multiple microbial populations simultaneously, [...]”.*

**2. According to their findings, there is little correlation between mRNA expression changes and**
**the corresponding alterations on the protein level and it is thus “nearly impossible to predict**
**the level of a given protein based on the level of the corresponding transcript” (see lines 184-**
**186). Put provocatively, this raises the question as to why at all (meta-)transcriptomic**

experiments should be conducted. This is highly relevant for many researchers as RNA-seq
is widely used and the authors should therefore provide here some guidelines as to when
RNA-seq might still provide functional implications. (Or, in case they generally discourage
from using RNA-seq for functional bacterial analyses, they should phrase it as such.)

**RESPONSE:** We believe that the use of RNA-seq is extremely relevant in biology, according to a
meaningful time spacing of the sampling and careful analysis. Indeed, transcript levels store the “recent
history” (up to minutes) of the cell and the community at large, whilst the proteins usually remain viable
longer (up to hours). Moreover, the correlation results at **Line 205** concern individual transcripts and
proteins correlated over time; while in the rest of the paper we show how it is more meaningful it is to
analyze the relationship between proteome and transcriptome at each time point. Regardless, we agree
with the reviewer that our results could raise questions as to which omic technology (transcriptomics
or proteomics) should be applied to assess community function, and have added additional text to
highlight that both have merit and should be considered (if possible):

**Line 389:** “*The observed discrepancy between RNA and protein levels of a given gene within the SEM1b*
*consortium (i.e. Fig. 2b) could raise questions as to which omic technology (transcriptomics or*
*proteomics) should be applied to assess community function. We would argue that both technologies*
*have merit and if possible, should be applied to the same sample(s), given that transcript levels store*
*the “recent history” (up to minutes) of a cell and/or the community at large, whilst proteins usually*
*remain viable much longer (up to hours) and can result in a different interpretation of function.”*

**3. Some parts would benefit from a more detailed experimental description. For example, the**
**authors should provide more experimental details of their metatranscriptomics analysis.**
**For example, line 449 reads “After purification, residual DNA, free nucleotides and small**
**RNAs were removed.” But it is not explained HOW this was achieved. Likewise, line 450:**
**“Samples were treated to enrich for mRNAs (...)” Here again, how this was done is not**
**mentioned. Further, I’d appreciate if the authors compiled a supplementary table with the**
**mapping statistics of the metatranscriptomics data (number of reads/sample; percentage of**
**mapped vs. unmapped reads/sample; distribution of the mapped reads to their respective**
**source genomes; etc.). This would also help the reader to obtain an idea as to how the**
**relative composition of the consortium changes over time (or if it remains unchanged).**

The overall experimental design is still unclear to me: In lines 430-432 it is stated that “The
time series analyses consisted of metabolomics, metaproteomics and metatranscriptomics
over nine time points (...) in triplicate”. However, reading on it sounds like not all time
points of this timecourse were analyzed by all three omics approaches. Could the authors
please clarify? In general, a supplementary figure showing a scheme of the samples taken
and indicating with which omic method they were analyzed would help the reader to better
appreciate their study.

Also in the methods section, the term “as previously described” should be avoided; rather,
the experiment should be fully described in the current manuscript (I believe this is
anyways an author guideline given by the journal).

**RESPONSE:** In response to this comment, we had made several adjustments and included much more
additional detail to how our experiments were performed. For example:

**sRNA removal, Line 506:** “After purification, residual DNA was removed using the Turbo DNA Free
kit following manufacturer’s instructions. Free nucleotides and small RNAs such as tRNAs were
cleaned off with a lithium chloride precipitation solution according to Thermo Fisher Scientific’s
recommendations ([https://www.thermofisher.com/be/en/home/references/ambion-tech-support/rna-](https://www.thermofisher.com/be/en/home/references/ambion-tech-support/rna-isolation/general-articles/the-use-of-licl-precipitation-for-rna-purification.html)
[isolation/general-articles/the-use-of-licl-precipitation-for-rna-purification.html](https://www.thermofisher.com/be/en/home/references/ambion-tech-support/rna-isolation/general-articles/the-use-of-licl-precipitation-for-rna-purification.html)) Briefly, one volume
of cold 5M LiCl solution was added to the samples, mixed well and incubated at –20°C for 30 minutes.
Samples were centrifuged at maximum speed for 30 minutes at 4°C. The supernatants were discarded
and the pellets were washed with 70% ethanol prior to be resuspended in 16µl of RNase-free water.”

**mRNA enrichment, Line 514:** “To reduce the amount of rRNAs, the samples were treated to enrich
for mRNAs using the MICROBExpress kit (Applied Biosystems, USA). The successful rRNA depletion
was confirmed by analyzing both pre- and post-treated samples on a 2100 bioanalyzer instrument. The
enriched mRNA was amplified with the MessageAmp II-Bacteria Kit (Applied Biosystems, USA)
following manufacturer’s instruction and sent for sequencing at the NSC (Oslo, Norway).”

**MT reads table:** We assembled the new **Supplementary Dataset 6** to show the MT reads throughout
the analysis. The table lists, per sample:

- 1. The number of starting reads;
2. The number of filtered reads (after quality, length, tRNA and rRNA filtering);

- 3. The fraction of filtered reads respect to the starting reads;
- 4. The number of filtered biological reads (i.e. without the internal standard);
- 5. The fraction of filtered biological reads respect to the total of filtered reads per sample;
- 6. The number of filtered internal standard reads;
- 7. The number of filtered biological reads mapped on the transcript dataset;
- 8. The fraction of filtered biological reads mapped respect to the filtered biological reads.

In addition, our MT quantification pipeline contains mainly three steps: i) pseudoalignment with
kallisto in which multiple alignments per read are allowed, ii) estimation using mmseq of per-ORF MT
quantification as Reads Per Kilobase Million (RPKM), iii) estimation via mmcollapse of per-ORFG
(ORF Group) MT quantification again as Reads Per Kilobase Million (RPKM). Therefore, as a proxy
of the number of reads mapped per genome, we hope that the genome-wise sum of the RPKM values
from the output of step ii is sufficient. We provide these values in the new **Supplementary Dataset 7**.
The values are presented in pairs of columns listing the sum of per-sample and per-genome RPKM
values, alongside to the fraction respect to the total per-sample RPKM values.

**Experimental design figure**: To improve clarity of our experimental design, we have generated a new
**Figure 1** (illustrated above for **Reviewer#1_Q15**) to illustrate the sampling scheme. Moreover, the
figure integrates the representation of the growth curve of the community (**Reviewer#1_Q15**) and the
explanation of the timeline and metabolism (**Reviewer#2_Q4**).

**“As previously described”**: As the reviewer suggests, we have removed the expression “as previously
described” (and its variations) and substituted it with the exhaustive description of the methods we
used:

**Line 76**: “*We previously genomically reconstructed and resolved the SEM1b community, retrieving 11*
*metagenome assembled genomes (MAGs) as well as two isolate genomes (see Material and Methods)¹⁰*
*[...]*”

**Line 477**: “*Non-invasive DNA extraction method was used to extract high molecular weight DNA as*
*previously described in Kunath et al.⁴⁷. A cell pellet was produced by centrifugation of 2ml of samples*

at 14, 000 x g for 5 minutes. Pellet was resuspended in 1ml of RBB+C buffer (500mM NaCl, 50mM
Tris-HCl; 50mM EDTA, 4% SDS) and incubated for 20 minutes at 70°C. NaCl solution was used to
reach 0.7M and 1:10 volume of CTAB buffer was added before an additional incubation at 70°C for
10 minutes. An equal volume of Chloroform is then added and centrifuged at 14,000 rcf for 15 minutes.
The aqueous phase was retrieved and an equal volume of Phenol:Chloroform:Isoamylalcohol
(25:24:1) is added and centrifuged at 14,000 rcf for 15 minutes. The aqueous phase was retrieved one
more time and 2 volumes of 95% ethanol were added and gently mixed until the DNA spooled and
could be transferred with a sterile loop to a tube containing 200µl of 70% ethanol. After centrifugation
at 14,000 rcf for 2 minutes, the supernatant was discarded, and the pellet air-dried prior being
resuspended into 30µl of TE buffer (pH 8.0).”

**Line 491:** “The reads were 3’-trimmed (Phred<20, length>100) with cutadapt⁴⁹ and filtered using
FASTX-Toolkit (http://hannonlab.cshl.edu/fastx_toolkit/) to retain the reads with Phred>30 on at least
90% of their length. The reads were mapped (ID=100%) on two *Coprothermobacter proteolyticus*
isolates from SEM1b using the Burrows-Wheeler Aligner with maximal exact matches (BWA-MEM)⁵⁰.
The remaining reads were assembled with MetaSpades v 3.10.0 (k-mers: 21, 33, 55, 77)⁵¹ and the
contigs binned with Metabat v0.26.3 (in “very sensitive mode”). The contigs were also uploaded to the
Microbial Genomes and Microbiomes⁵² system for gene prediction and annotation.”

**Line 537:** “Proteins were extracted from t1 to t8 in triplicate. From each sample, 30ml of culture
containing cells and substrate was centrifuged at 500x g for 5 minutes to pellet the substrate.”

**Line 523:** “The resulting sequences were checked for overrepresented features with FastQC
(www.bioinformatics.babraham.ac.uk/projects/fastqc/); features and low qualities (Phred<20) ends
were trimmed using Trimmomatic⁵³ v.0.36. The reads were then filtered using an average Phred>20
in a 10nt window and a minimum length of 100 nt. The rRNA and tRNA reads were removed as using
SortMeRNA⁵⁴ v2.1b.”

**Line 597:** “SCFAs were then measured at 210 nm using a Dionex 3000 HPLC with a Zorbax Eclipse
Plus C18 column from Agilent Technologies (150 x 2.1mm (3.5mm particles)) and operated at 40°C.

*The VFAs were eluted isocratically with 100% methanol and 2.5 mM H2SO4 at 0.3 ml per minute flow*
*rate.”*

**Minor remarks:**

**4. Line 89: Change “algorithms has” to “algorithms have”.**

**RESPONSE:** Corrected as suggested.

**5. Line 158: “RNA is regulated by post-translational modifications of the RNA molecule” **

**Do the authors mean post-TRANSCRIPTIONAL modifications?**

**RESPONSE:** Corrected as suggested.

**6. Line 201: “start at values between 0.6 and 0.8 at 13 hours”  Please rephrase as there are**

**clearly values outside this range in Fig. 1d (also for non-TEPI2 MAGs).**

**RESPONSE:** Rephrased as “*start at values above 0.5 at 13 hours (t2)*”.

**7. Lines 239-240: “Notably from Fig. 2, it is clear that the proteins from the transporters are**

**almost never found in the samples, even if the respective RNAs are abundant.”  As far as**

**I understand, the discrepancy between RNA and protein level detection cannot be deduced**

**from Fig. 2.**

**RESPONSE:** The wording was misleading and we changed it to “*even if the respective RNAs are*

*present in the dataset*”. Indeed, Fig 2 takes into account only RNAs and proteins that are present in

the dataset (i.e. that passed the preprocessing threshold regarding expression), regardless of their

numerical expression.

**8. Fig. 3 b-d: The units for the values plotted on the y-axes are missing (also not mentioned in**

**the corresponding figure legend).**

**RESPONSE:** The y-axis for gene expression plots Fig. 3 depicts scale of log10-transformed

transcript molecules per sample. We used the same scale as in panel a, and to improve clarity have

added “*For panels b-d, RNA expression uses same scale as panel a*” to the legend of panels b-d.

**9. Line 336: “in bacteria is believed to occur predominantly via transcription control (...)” **
**The authors may want to rephrase this. This concept has been overhauled in the past**
**decade, realizing the widespread post-transcriptional control mechanisms – brought about**
**by regulatory, noncoding RNAs – across the bacterial phylogenic tree.**

**RESPONSE:** In response to this comment, we have changed the sentence to be more exhaustive and
neutral:

**Line 358:** *“The control of protein levels in bacteria is believed to occur via transcription control,*
*“control by dilution”⁴² (dispersal of proteins via subsequent cell divisions), sRNA activity⁴³, and rarely*
*by protein degradation⁴⁴.”*

**10. Line 511 (and elsewhere): Please define what “Nt” refers to in this context.**

**RESPONSE:** We added “expressed in nucleotide length”.

**11. Suppl. Fig. S1: The x-axis for the metabolomics bar chart lacks any values.**

**RESPONSE:** In Suppl. Fig. S1 the scale along the x axis is the same in the three panel. The last one
does not reach the 1000-counts tick. We have included the “500” ticks on the axis to depict this.

**REFERENCES**

**Cox 2014:** Cox J, *et al.* Accurate proteome-wide label-free quantification by delayed normalization
and maximal peptide ratio extraction, termed MaxLFQ. *Mol Cell Proteomics*. 13(9):2513-2526
(2014).

**Kunath 2019:** Kunath, B. J. *et al.* From proteins to polysaccharides: lifestyle and genetic evolution
of *Coprothermobacter proteolyticus*. *ISME J*. 13, 603–617 (2019).

**Lim 2019:** Lim *et al.* Evaluating False Transfer Rates from the Match-between-Runs Algorithm with
a Two-Proteome Model. *J Proteome*. 18(11):4020-4026 (2019).

**Liu 2016:** Liu Y., *et al.* On the Dependency of Cellular Protein Levels on mRNA Abundance. *Cell*.
165(3):535-550 (2016).

**Palkova 2004:** Palkova Multicellular microorganisms: laboratory versus nature. *EMBO Rep*. 5: 470-
476 (2004)

**Peng 2012:** Peng M. *et al.* Protease bias in absolute protein quantitation. *Nat Methods*. 9(6):524-525
(2012).

**Taniguchi 2010:** Taniguchi Y., *et al.* Quantifying *E. coli* proteome and transcriptome with single-
molecule sensitivity in single cells [published correction appears in *Science*. 28;334(6055):453 (2011)

**Thomason 2010:** Thomason M.K., *et al.* Bacterial antisense RNAs: How many are there and what are
they doing? *Annu Rev Genet*. 44: 167–188 (2010).

**Wiśniewski 2014:** Wiśniewski J.R. & Rakus D. Multi-enzyme digestion FASP and the 'Total Protein
Approach'-based absolute quantification of the *Escherichia coli* proteome. *J Proteomics*. 109:322-331
(2014).

REVIEWERS' COMMENTS:

Reviewer #1 (Remarks to the Author):

The authors have addressed all my comments to my satisfaction.

Excellent work!

Manuel Kleiner

Reviewer #2 (Remarks to the Author):

In this revision the authors addressed most of my major concerns and the clarity of the manuscript improved as well. I believe that the manuscript is fit for publication (perhaps after shortening a bit).

One point that remains is the potential biases that result from using only singletons for downstream analysis. I believe that readers would benefit from a very brief discussion of possible limitations that result from this analysis.

Minor:

Figure 2 axis labels and tick labels: should be consistent in font and size. Also, I believe it is easier to read a number or exponent (i.e., 10) rather than its mathematical representation (i.e., 1×10^2). Also please state in the legend the properties of the polynomial fit in 2d.

Reviewer #3 (Remarks to the Author):

My previous comments were all addressed satisfactorily. Apart from the below, I have nothing else to object.

- Line 354: “liked” should be changed to “linked”.
- Ref 43 might be replaced by a more recent review (e.g. PMID 26296935).

**REVIEWERS' COMMENTS:**

**Reviewer #1 (Remarks to the Author):**

**The authors have addressed all my comments to my satisfaction.**

**Excellent work!**

**RESPONSE:** We thank the referee for the insightful revision.

**Manuel Kleiner**

**Reviewer #2 (Remarks to the Author):**

**In this revision the authors addressed most of my major concerns and the clarity of the manuscript**
**improved as well. I believe that the manuscript is fit for publication (perhaps after shortening a bit).**

**RESPONSE:** We thank the reviewer for their work in improving the manuscript, we hope that the following
answers will improve it further.

**One point that remains is the potential biases that result from using only singletons for downstream**
**analysis. I believe that readers would benefit from a very brief discussion of possible limitations that**
**result from this analysis.**

**RESPONSE:** We clarified the limitations and the exclusion of some populations from the analysis stemming
from those limitations in Line 634 by adding the sentences: *“The subset may suffer marginally from a loss in*
*data points (ORFs), however the genomes in which this phenomenon had a larger impact (COPR2-3, BWF2A*
*and SW3C) were not used to estimate numerical properties such as protein-to-RNA ratios and k values. In*
*addition, the impact of data loss for the aforementioned MAGs/strains was illustrated in Supplementary Figure*
*2 and did not outline any clear distribution that was opposing the observations made for the MAGs used in this*
*study.”*

**Minor:**

**Figure 2 axis labels and tick labels: should be consistent in font and size. Also, I believe it is easier to**
**read a number or exponent (i.e., 10) rather than its mathematical representation (i.e., 1+e02). Also**
**please state in the legend the properties of the polynomial fit in 2d.**

**RESPONSE:** We made the size and the font uniform in the whole figure, as requested by the referee #2. The
third-grade polynomial fit allows up to two bends to the curve. This information has been added to the Fig. 2d
legend as requested.

**Reviewer #3 (Remarks to the Author):**

**My previous comments were all addressed satisfactorily. Apart from the below, I have nothing else to**
**object.**

**RESPONSE:** We thank the referee for the work on the manuscript and the suggestions presented.

• **Line 354: “liked” should be changed to “linked”.**

**RESPONSE:** Changed as requested.

• **Ref 43 might be replaced by a more recent review (e.g. PMID 26296935).**

**RESPONSE:** Changed as requested.
